# Combined effects of gliding-arc plasma and C-phycocyanin on antioxidant activity and shelf-life extension of rainbow trout (*Oncorhynchus mykiss*) fillets

Maedehsadat Seyedalangi[1], Amir Hossein Sari[2], Bahareh Nowruzi[3]*, Seyed Amir Ali Anvar[4]

**1** Plasma Physics Research Center, Science and Research Branch, Islamic Azad University, Tehran, Iran, **2** Department of Physics, Science and Research Branch, Islamic Azad University, Tehran, Iran, **3** Department of Biology, SR.C., Islamic Azad University, Tehran, Iran, **4** Department of Veterinary Hygine, SR.C., Islamic Azad University, Tehran, IRAN

* bahare77biol@gmail.com, baharehnowruzi77@iau.ac.ir

## Abstract

The study aimed to evaluate the impact of gliding arc plasma (GAP) treatment and phycocyanin pigment (PCP) on the antioxidant activity and refrigerated shelf-life extension of *Oncorhynchus mykiss* (rainbow trout) fillets (OMR) during 18 days of storage at $4 \pm 0.1$ °C. The combined treatment of GAP and PCP significantly inhibited microbial growth, with the total viable count in the control sample (C) increasing from 4.39 to 12.08 log CFU/g by day 18, while the combined treatment (P5+PC) increased only from 3.50 to 9.23 log CFU/g, extending the shelf life from 3 to 12 days. Antioxidant activities (DPPH, ABTS, and FRAP) were markedly higher in treated samples, as DPPH values changed only slightly (from 4.20 to 5.23) compared to the control (from 10.91 to 36.41), and FRAP activity decreased less sharply (35.88 to 25.30 vs. 20.01 to 11.75). Moreover, the total volatile nitrogen (TVN) value in the control rose from 8.77 to 53.95 mg N/100 g, exceeding the international limit (30 mg N/100 g) by day 9, whereas the combined treatment remained below this limit (29.40 mg N/100 g) even on day 18. The n-3/n-6 fatty acid ratio slightly decreased (0.14 to 0.11) but remained within an acceptable nutritional range. Overall, the results indicate that the combined GAP and PCP treatment is an effective and promising approach to extend the shelf life of OMR for more than 12 days under refrigerated conditions while maintaining better antioxidant stability.

## 1. Introduction

Driven by increasing consumer demand for fresh and minimally processed foods—particularly seafood—producers are progressively adopting more sustainable practices. rainbow trout (*Oncorhynchus mykiss*) is recognized as a species of global

**Data availability statement:** All relevant data are within the manuscript and its Supporting Information files.

**Funding:** The author(s) received no specific funding for this work.

**Competing interests:** The authors have declared that no competing interests exist.

ecological and economic significance [1]. Although food additives are widely used in food manufacturing, rising concerns about their safety and potential health risks have prompted a shift in industry practices [2].

Consequently, there is growing interest in natural additives and innovative non-thermal preservation technologies, such as cold plasma, to produce safer and higher-quality foods with minimal processing [3].

The application of natural pigments in food products has gained substantial research attention, with phycocyanin pigment (PCP) emerging as a particularly promising candidate. This vibrant blue pigment, derived mainly from cyanobacteria such as *Spirulina platensis*, functions not only as a natural colorant but also as a bioactive compound with demonstrated antioxidant, anti-inflammatory, and anti-carcinogenic properties. [4].

Low-temperature plasma (cold plasma) processing represents an emerging non-thermal technology in food science. It offers a powerful alternative to conventional thermal preservation methods, particularly for heat-sensitive products such as fresh seafood and produce [5]. Among cold plasma technologies, gliding arc plasma (GAP) has shown considerable potential in preserving fresh foods [6]. A major advantage of GAP is its ability to inactivate pathogenic and spoilage microorganisms on food surfaces, thereby enhancing food safety and extending shelf-life by reducing the risk of foodborne illness [7]. As a non-thermal technology, GAP prolongs the shelf-life of perishable commodities by generating reactive oxygen and nitrogen species (RONS), that inhibit microbial growth while preserving sensory and nutritional qualities. Although previous studies have established the effectiveness of GAP in food preservation there is limited research on its synergistic application with natural bioactive compounds such as PCP.

Therefore, the present study aims to address this gap by systematically evaluating the combined effects of GAP (2 min and 5 min) and PCP on microbial spoilage, antioxidant activity, and the refrigerated shelf life of rainbow trout fillets.

## 2. Materials and methods

### 2.1 Materials

Live rainbow trout (*Oncorhynchus mykiss*) were purchased from a local market in Tehran, Iran. The phycocyanin pigment was extracted from laboratory-cultivated *Spirulina platensis* using a freezing–thawing and centrifugation process, followed by purification and lyophilization.

All chemicals and reagents used were of analytical grade and obtained from Iranian suppliers, including trichloroacetic acid, thiobarbituric acid, chloroform, methanol, ethanol, glacial acetic acid, sodium hydroxide, hydrochloric acid, and phosphoric acid.

Microbiological media, such as Plate Count Agar (PCA), Baird–Parker Agar (BPA), Selenite Cystine Broth (SCB), Xylose Lysine Deoxycholate (XLD) agar, MacConkey Agar (MCA), de Man–Rogosa–Sharpe (MRS) agar, Violet Red Bile Glucose Agar (VRBG), were purchased from Liofilchem (Roseto degli Abruzzi, Italy).

All other materials and solvents were prepared using distilled water and standard analytical-grade reagents.

## 2.2 PCP preparation and purity assessment

PCP was extracted following the cultivation procedure of *Spirulina platensis* cyanobacteria as described by Pan et al. [8]. After purification, cultures were maintained in Zarrouk's liquid medium, and citric acid was added as a preservative to enhance pigment stability. The minimum inhibitory concentration (MIC) of PCP was determined using the standard broth dilution method. The purity of the pigment was assessed spectrophotometrically according to the following formula (Eq. 1):

$$\text{The purity of PC} = \frac{A620}{A280} \tag{1}$$

Where A620 corresponds to the absorbance of PCP and *A280* corresponds to the absorbance of total proteins.

## 2.3 Sample preparation and ethical considerations

Fresh rainbow trout fillets were obtained from a local market in Tehran, Iran. Fish were washed under a continuous flow of tap water, and the head and gastrointestinal tract were removed. The fillets were then cut into standardized dimensions of $5 \times 5 \times 1$ cm. A total of 200 fillets were used in this study. To ensure precision, all analyses were performed in triplicate [9]. Half of the fillets were immersed in a PCP solution with a concentration of 0.065 mg/mL for 1 h at 4 °C, as determined in preliminary assays [10]. Following inoculation, all fillets were placed in sterile Petri dishes for GAP treatment. GAP exposure was conducted using a Plasmatech-17A device (Kavosh Yaran Fan Poya Company, Tehran, Iran) (Fig 1). The system was powered by an AC supply (output power 1.5 kW, 50 Hz, voltage ~10 kV). The GAP flame was directed at a fixed

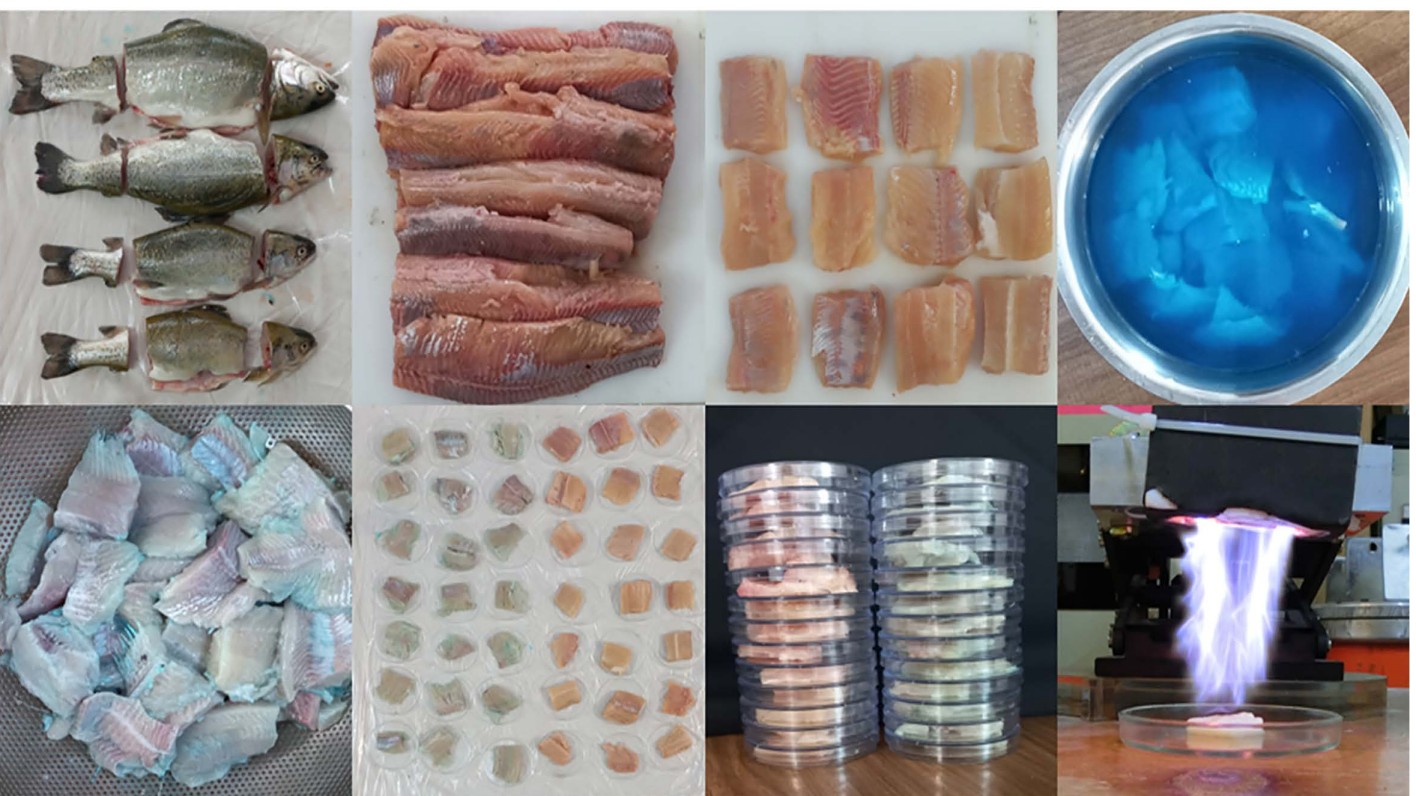

**Fig 1. Sequential steps involved in the preparation and plasma treatment of rainbow trout fillets.**

distance of 5 cm from the nozzle to the fillet surface. Treatments were performed for either 2 min or 5 min under constant power in the presence of air at room temperature.

To facilitate clarity, treatment groups were abbreviated as follows: **C** (control sample without plasma treatment and without phycocyanin pigment), **PC-P** (sample treated with phycocyanin pigment but without plasma), **P2-PC** (plasma-treated sample for 2 minutes without phycocyanin pigment), **P5-PC** (plasma-treated sample for 5 minutes without phycocyanin pigment), **P2+PC** (sample treated with plasma for 2 minutes and phycocyanin pigment), and **P5+PC** (sample treated with plasma for 5 minutes and phycocyanin pigment) (Fig 2).

All methods were carried out in accordance with relevant guidelines and regulations. Authors don't do any experiments on humans and/or the use of human tissue samples. All experimental protocols and panelists involved in the study were approved by ethics committee of Tehran medical sciences, Islamic Azad University, Tehran, Iran, with the reference number: IR. IAU. SRB.REC.1404.110.

## 2.4 Microbiological analyses

The microbial groups commonly used as indicators of spoilage in food products include total viable count)TVC(and psychrotrophic bacterial counts (PTC) [11].TVC and PTC were enumerated on PCA incubated at 30 °C for 48 h and 7 °C for 10 days, respectively [12]. Coagulase-positive *Staphylococcus* spp. were determined on BPA supplemented with egg yolk tellurite emulsion and incubated at 37 °C for 48 h [13]. Pathogenic bacteria were also assessed: *Salmonella enterica* was detected using selective enrichment in SCB followed by plating on XLD agar, and *Escherichia coli*) *E. coli* (was enumerated on MCA incubated at 37 °C for 24 h [14]. *Lactobacillus* spp. were cultured on MRS agar under anaerobic conditions at 30 °C for 72 h [15]. Members of the family *Enterobacteriaceae* were quantified on VRBG agar incubated at 37 °C for 24 h [16]. All microbial counts were expressed as the logarithm of colony-forming units per gram of sample (log CFU/g).

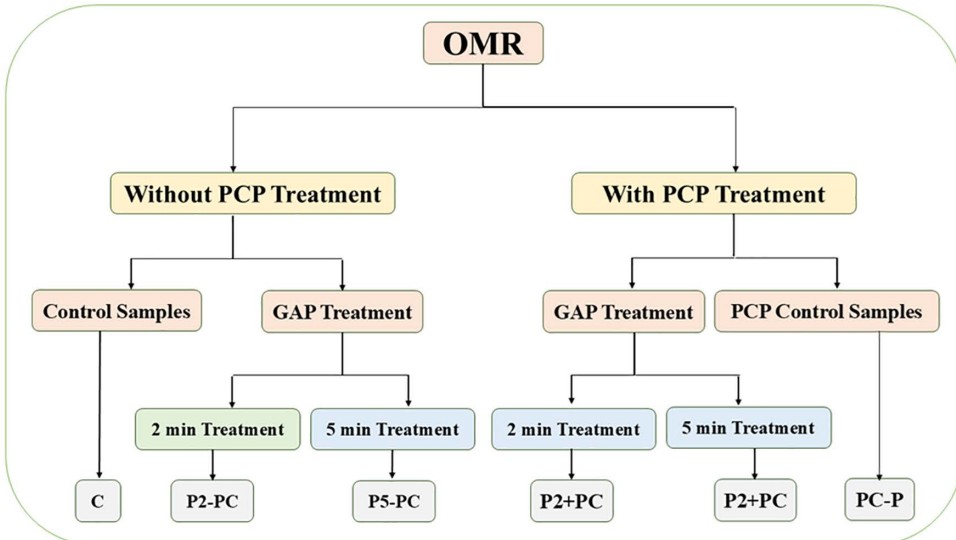

**Fig 2. Schematic representation of the experimental design for the treatment of rainbow trout fillets with gliding arc plasma and phycocyanin pigment.** The phycocyanin (PCP) concentration was fixed at 0.065 mg/mL. Half of the samples were pretreated by immersion in the phycocyanin solution for 1 h at 4 °C prior to plasma exposure. The experimental groups were as follows: C (control sample without plasma or pigment), PC-P (sample treated with phycocyanin pigment only), P2-PC (plasma-treated sample for 2 min without pigment), P5-PC (plasma-treated sample for 5 min without pigment), P2+PC (sample treated with plasma for 2 min and phycocyanin pigment), and P5+PC (sample treated with plasma for 5 min and phycocyanin pigment).

## 2.5 Physicochemical properties

The physicochemical properties of fillets were evaluated through standard assays, including pH, titratable acidity, peroxide value (PV), thiobarbituric acid reactive substances (TBARS), total volatile nitrogen (TVN), trimethylamine (TMA), fatty acid profile, and instrumental color analysis [17]. The pH of the samples was determined in a 1:10 (w/v) homogenate using a calibrated digital pH meter. Titratable acidity was measured by titration with 0.1 N NaOH and expressed as lactic acid equivalent (% w/v) [18]. Lipid oxidation was assessed by quantifying the PV using the iodometric titration method [19]. Secondary lipid oxidation products were determined by the TBARS assay, and malondialdehyde (MDA) concentration was calculated according to Eq. (2), based on spectrophotometric absorbance readings at 532, 600, and 450 nm [9].

Final values were expressed as mg MDA/kg sample according to Eq. (3) [20,21]:

$$c \left( \mu molL^{-1} \right) = 6.45 \ (A_{532} - A_{600}) - 0.56 \times A_{450} \tag{2}$$

$$MDA \ concentration \ \left( \mu molKg^{-1} \ dry \ weight \right) = \frac{c \times v_r \times v_e}{v_d \times m \times 1000} \tag{3}$$

Protein degradation was quantified by measuring TVN and TMA using the micro-Kjeldahl distillation method. Results were expressed as mg N/100 g sample for TVN and mg TMA-N/100 g sample for TMA, both of which serve as widely accepted indicators of spoilage in fish [22].

To evaluate lipid nutritional quality, the fatty acid profile (FAP) was determined following lipid extraction, conversion to fatty acid methyl esters, and analysis by gas chromatography. Results were expressed as the percentage of total fatty acids, and the n-3/n-6 ratio was calculated. Analyses were carried out on four representative groups: C, PC-P, P5-PC, and P5 + PC [23].

Color parameters were measured using a Hunter Lab Mini scan/EZ colorimeter under a D65 illuminant with a 10° standard observer. Lightness (L*), redness/greenness (a*), and yellowness/blueness (b*) values were recorded. Overall color difference (ΔE) was calculated relative to day 1 using Eq. (4), where ΔL*, Δa*, and Δb* represent the respective changes in color coordinates [22]:

$$\Delta E = \sqrt{(\Delta L^*)^2 + (\Delta a^*)^2 + (\Delta b^*)^2} \tag{4}$$

## 2.6 Antioxidant evaluation

The antioxidant capacity of the fillets was assessed using three standard assays: 2,2-diphenyl-1-picrylhydrazyl (DPPH) radical scavenging activity, ferric reducing antioxidant power (FRAP), and ABTS radical cation decolorization. The radical scavenging activity against DPPH was measured using ethanol as the solvent, and absorbance was read at 517 nm. Radical scavenging activity (%) was calculated using Eq. (5), in which $A_0$ represents the absorbance of the DPPH solution without the sample, and $A_s$ denotes the absorbance of the DPPH solution in the presence of the sample after incubation [20].

$$DPPH \ scavenging \ rate \ (\%) = \frac{A0 - As}{A0} \times 100 \tag{5}$$

FRAP and ABTS radical scavenging activity were determined spectrophotometrically according to standard methods, and the result were expressed as the percentage of radical inhibition and $Fe^{2+}$ equivalents per gram of sample [24].

$$\frac{\text{Absorbance Control} - \text{Absorbance sample}}{\text{Absorbance control}} \times 100 \qquad (6)$$

## 2.7 Sensory evaluation

Sensory quality of the fillets was evaluated by a trained panel of ten members (five males and five females). The attributes assessed included color, odor, texture, and overall acceptability. A 5-point hedonic scale was employed, where 1 = extremely good, 2 = good, 3 = acceptable, 4 = poor, and 5 = extremely unacceptable. Each sample was coded with random three-digit numbers and presented to panelists in a randomized order under controlled laboratory conditions to minimize bias. Water was provided between evaluations to cleanse the palate [24].

## 2.8 Statistical analysis

3.All experiments were performed in triplicate during 18 days of refrigerated storage at 4 °C. Data were analyzed using SPSS software. One-way analysis of variance (ANOVA) was applied to evaluate significant differences among treatments and storage times at a 95% confidence level ($p < 0.05$). When ANOVA indicated significance, Tukey's post hoc test was performed for pairwise comparisons of means. All data measurements were performed in triplicate, and results were expressed as mean ± standard error of the mean (SEM). Significant differences among treatments and storage times were considered at $p < 0.05$. Graphical representations of the data were generated using Microsoft Excel. [25].

# Results

## 3.1 Concentration and purity of PCP

The concentration and purity of PCP were evaluated before and after dialysis. The initial concentration and purity values were 0.238 mg/mL and 0.063, respectively, which increased to 0.251 mg/mL and 0.065 after dialysis. The purity of PCP was further confirmed by spectrophotometric analysis, where a distinct absorption peak at 621.9 nm was observed S1 Fig, indicating the presence of PCP with a high degree of purity.

## 3.2 Microbiological analyses

**3.2.1 TVC and PTC.** As shown in Table 1, TVC of all samples increased progressively throughout the 18 days of refrigerated storage. In the control samples, TVC rose significantly from 4.39 log CFU/g on day 1 to 12.08 log CFU/g on day 18 ($p < 0.05$), exceeding the acceptable spoilage limit of 6 log CFU/g on day 3. In the P5-P control samples, microbial growth was slower, with TVC values increasing from 3.97 on day 1 to 10.90 log CFU/g on day 18 ($p < 0.05$), and the spoilage limit was reached on day 9. The PC-P group showed intermediate inhibition, with TVC rising from 4.12 to 11.77 log CFU/g ($p < 0.05$), exceeding the acceptable limit on day 6. The combined treatment (P5 + PC) demonstrated the most pronounced reduction in microbial load, with TVC increasing from 3.50 to 9.23 log CFU/g by day 18 ($p < 0.05$). In this group, the spoilage threshold was not reached until day 12, confirming superior microbial control among all treatments.

As shown in S1 Table, the PTC followed a similar increasing trend during storage. In the control samples, PTC increased from 3.32 log CFU/g on day 1 to 10.39 log CFU/g on day 18 ($p < 0.05$). In the P5-PC group, PTC values rose from 2.95 to 8.70 log CFU/g, while the PC-P samples showed an increase from 3.14 to 9.77 log CFU/g ($p < 0.05$). The P5 + PC treatment exhibited the lowest PTC throughout the storage period, increasing from 2.70 log CFU/g on day 1 to 7.17 log CFU/g on day 18 ($p < 0.05$).

**3.2.2 *Staphylococcal coagulase-positive* bacteria count.** As shown in Table 2, the counts of *Staphylococcal coagulase-positive* bacteria were initially undetectable (0 log CFU/g) in all samples on day 1, with no significant differences among treatments ($p > 0.05$). During storage, bacterial counts increased progressively in all groups. In the

**Table 1. Mean TVC of *Oncorhynchus mykiss* fillets treated with GAP and PCP during storage at 4°C for 18 days.**

| TVC | Day1 | Day3 | Day6 | Day9 | Day12 | Day15 | Day18 |
|---|---|---|---|---|---|---|---|
| C | 4.39±0.0078(a) (A) | 6.37±0.0083(a) (B) | 7.36±0.0067(a) (C) | 8.24±0.0209(a) (D) | 9.34±0.0069(a) (E) | 11.36±0.0166(a) (F) | 12.08±0.0128(a) (G) |
| P2-PC | 4.09±0.0261(b) (A) | 6.01±0.0205(b) (B) | 6.77±0.0519(b) (C) | 7.32±0.0063(b) (D) | 8.25±0.0160(b) (E) | 10.87±0.0420(b) (F) | 11.33±0.0179(b) (G) |
| P5-PC | 3.97±0.0178(c) (A) | 5.88±0.0344(c) (B) | 5.95±0.0102(c) (C) | 7.09±0.0144(c) (D) | 7.40±0.0062(c) (E) | 10.39±0.0118(c) (F) | 10.90±0.0371(c) (G) |
| PC-P | 4.12±0.0069(b) (A) | 5.57±0.0472(d) (B) | 6.87±0.0090(d) (C) | 7.92±0.0185(d) (D) | 9.22±0.0145(d) (E) | 11.13±0.0113(d) (F) | 11.77±0.0532(d) (G) |
| P2+PC | 3.82±0.0264(d) (A) | 4.95±0.0345(e) (B) | 5.42±0.0033(e) (C) | 6.20±0.0124(e) (D) | 6.95±0.0175(e) (E) | 9.27±0.0198(e) (F) | 10.09±0.0116(e) (G) |
| P5+PC | 3.50±0.0454(e) (A) | 4.54±0.0739(f) (B) | 5.12±0.0184(f) (C) | 5.84±0.0126(f) (D) | 6.51±0.0490(f) (E) | 8.38±0.0059(f) (F) | 9.23±0.0089(f) (G) |

C: control sample (without plasma treatment and phycocyanin pigment); PC-P: sample treated with phycocyanin pigment but without plasma; P2-PC: plasma-treated sample for 2 min without phycocyanin pigment; P5-PC: plasma-treated sample for 5 min without phycocyanin pigment; P2+PC: plasma-treated sample for 2 min with phycocyanin pigment; P5+PC: plasma-treated sample for 5 min with phycocyanin pigment. Different small and capital letters indicate significant differences in the columns and rows, respectively ($p < 0.05$).All data are expressed as mean±SEM (n = 3). Data were analyzed using one-way ANOVA followed by Tukey's post hoc test ($p < 0.05$).

**Table 2. Mean *Staphylococcal coagulase-positive* bacteria of *Oncorhynchus mykiss* fillets treated with GAP and PCP during storage at 4°C for 18 days.**

| *Staphylococcal coagulase-positive* | Day1 | Day3 | Day6 | Day9 | Day12 | Day15 | Day18 |
|---|---|---|---|---|---|---|---|
| C | 0.00±0.0000(a) (A) | 3.38±0.0051(a) (B) | 4.40±0.0026(a) (C) | 5.36±0.0092(a) (D) | 6.39±0.0079(a) (E) | 7.39±0.0033(a) (F) | 8.36±0.0049(a) (G) |
| P2-PC | 0.00±0.0000(a) (A) | 3.25±0.0085(b) (B) | 3.93±0.0103(b) (C) | 5.17±0.0065(b) (D) | 6.17±0.0070(b) (E) | 7.06±0.0055(b) (F) | 8.10±0.0061(b) (G) |
| P5-PC | 0.00±0.0000(a) (A) | 3.00±0.0187(c) (B) | 3.73±0.0374(c) (C) | 4.93±0.0119(c) (D) | 5.81±0.0164(c) (E) | 6.80±0.0167(c) (F) | 7.42±0.0240(c) (G) |
| PC-P | 0.00±0.0000(a) (A) | 2.79±0.0226(c) (B) | 4.18±0.0042(b) (C) | 5.07±0.0169(d) (D) | 6.09±0.0074(d) (E) | 7.16±0.0079(d) (F) | 7.95±0.0156(b) (G) |
| P2+PC | 0.00±0.0000(a) (A) | 2.62±0.0240(d) (B) | 2.92±0.0109(d) (C) | 4.15±0.0108(e) (D) | 5.02±0.0060(e) (E) | 7.14±0.0055(e) (F) | 7.01±0.0062(d) (G) |
| P5+PC | 0.00±0.0000(a) (A) | 2.34±0.0229(e) (B) | 2.44±0.0501(e) (C) | 3.65±0.0333(f) (D) | 4.77±0.0193(f) (E) | 5.56±0.0284(f) (F) | 6.70±0.0210(e) (G) |

C: control sample (without plasma treatment and phycocyanin pigment); PC-P: sample treated with phycocyanin pigment but without plasma; P2-PC: plasma-treated sample for 2 min without phycocyanin pigment; P5-PC: plasma-treated sample for 5 min without phycocyanin pigment; P2+PC: plasma-treated sample for 2 min with phycocyanin pigment; P5+PC: plasma-treated sample for 5 min with phycocyanin pigment. Different small and capital letters indicate significant differences in the columns and rows, respectively ($p < 0.05$). All data are expressed as mean±SEM (n = 3). Data were analyzed using one-way ANOVA followed by Tukey's post hoc test ($p < 0.05$).

control samples, the population reached 8.36 log CFU/g by day 18 ($p < 0.05$). The PC-P and P5-PC treatments showed reduced growth compared with C, reaching 7.95 and 7.42 log CFU/g, respectively, on day 18 ($p < 0.05$). The lowest bacterial growth was observed in the P5+PC samples, which reached only 6.70 log CFU/g on day 18 ($p < 0.05$).

**3.2.3 *Salmonella* spp. and *Escherichia coli* count.** No *Salmonella* or *E. coli* were detected in any samples throughout the 18-day period.

**3.2.4 *Lactobacillus spp.* Count.** As shown in Table 3, *Lactobacillus spp.* counts were initially undetectable (0.00 log CFU/g) across all treatments on day 1, with no significant differences observed ($p > 0.05$). Over the 18-day storage

**Table 3. Mean *Lactobacillus spp.* bacteria of *Oncorhynchus mykiss* fillets treated with GAP and PCP during storage at 4°C for 18 days.**

| *Lactoba-cillus spp.* | Day1 | Day3 | Day6 | Day9 | Day12 | Day15 | Day18 |
|---|---|---|---|---|---|---|---|
| C | 0.00±0.0000(a) (A) | 3.35±0.0086(a) (B) | 5.39±0.0016(a) (C) | 6.37±0.0064(a) (D) | 7.38±0.0058(a) (E) | 8.33±0.0083(a) (F) | 10.40±0.0065(a) (G) |
| P2-PC | 0.00±0.0000(a) (A) | 3.07±0.0068(b) (B) | 5.07±0.0057(b) (C) | 6.13±0.0107(b) (D) | 6.82±0.0076(b) (E) | 8.15±0.0054(b) (F) | 9.23±0.0045(b) (G) |
| P5-PC | 0.00±0.0000(a) (A) | 2.92±0.0152(c) (B) | 4.45±0.0378(c) (C) | 5.77±0.0190(c) (D) | 6.39±0.0037(c) (E) | 8.00±0.0081(c) (F) | 8.72±0.0191(c) (G) |
| PC-P | 0.00±0.0000(a) (A) | 2.77±0.0194(c) (B) | 5.07±0.0117(b) (C) | 6.23±0.0067(d) (D) | 7.13±0.0056(d) (E) | 8.12±0.0142(d) (F) | 9.57±0.0491(b) (G) |
| P2+PC | 0.00±0.0000(a) (A) | 2.53±0.0153(d) (B) | 3.96±0.0087(d) (C) | 4.92±0.0090(e) (D) | 6.00±0.0140(e) (E) | 7.16±0.0165(e) (F) | 8.38±0.0074(d) (G) |
| P5+PC | 0.00±0.0000(a) (A) | 2.07±0.0732(e) (B) | 3.43±0.0427(e) (C) | 4.71±0.0098(f) (D) | 5.62±0.0336(f) (E) | 6.55±0.0257(f) (F) | 8.20±0.0118(e) (G) |

C: control sample (without plasma treatment and phycocyanin pigment); PC-P: sample treated with phycocyanin pigment but without plasma; P2-PC: plasma-treated sample for 2 min without phycocyanin pigment; P5-PC: plasma-treated sample for 5 min without phycocyanin pigment; P2+PC: plasma-treated sample for 2 min with phycocyanin pigment; P5+PC: plasma-treated sample for 5 min with phycocyanin pigment. Different small and capital letters indicate significant differences in the columns and rows, respectively ($p < 0.05$). All data are expressed as mean±SEM (n=3). Data were analyzed using one-way ANOVA followed by Tukey's post hoc test ($p < 0.05$).

period, a progressive increase in *Lactobacillus* populations was observed in all groups, though the extent of growth varied among treatments. The highest count was recorded in the control samples, reaching 8.36 log CFU/g by day 18 ($p < 0.05$). In contrast, P5-PC and PC-P treatments effectively delayed microbial proliferation, reaching 7.42 and 7.95 log CFU/g, respectively, on day 18 ($p < 0.05$). The combined treatment (P5+PC) demonstrated the most substantial inhibitory effect, maintaining the lowest *Lactobacillus spp.* load at 6.70 log CFU/g on day 18 ($p < 0.05$).

**3.2.5 *Enterobacteriaceae* counts.** As shown in Table 4, *Enterobacteriaceae* counts were initially undetectable (0.00 log CFU/g) across all samples on day 1, showing no significant differences ($p > 0.05$). During storage, bacterial growth

**Table 4. Mean *Enterobacteriaceae* bacteria of *Oncorhynchus mykiss* fillets treated with GAP and PCP during storage at 4°C for 18 days.**

| *Enterobac-teriaceae* | Day1 | Day3 | Day6 | Day9 | Day12 | Day15 | Day18 |
|---|---|---|---|---|---|---|---|
| C | 0.00±0.0000(a) (A) | 3.39±0.0060(a) (B) | 5.33±0.0042(a) (C) | 6.38±0.0038(a) (D) | 8.39±0.0036(a) (E) | 9.32±0.0049(a) (F) | 10.28±0.0093(a) (G) |
| P2-PC | 0.00±0.0000(a) (A) | 3.11±0.0171(b) (B) | 5.09±0.0063(b) (C) | 5.97±0.0188(b) (D) | 7.10±0.0127(b) (E) | 8.15±0.0044(b) (F) | 9.26±0.0035(b) (G) |
| P5-PC | 0.00±0.0000(a) (A) | 2.94±0.0123(b) (B) | 5.10±0.0030(b) (C) | 6.14±0.0127(c) (D) | 6.57±0.0267(c) (E) | 8.11±0.0078(c) (F) | 8.22±0.0060(b) (G) |
| PC-P | 0.00±0.0000(a) (A) | 2.69±0.0437(c) (B) | 5.29±0.0041(a) (C) | 6.14±0.0102(c) (D) | 7.62±0.0403(d) (E) | 8.67±0.0303(d) (F) | 9.83±0.0085(c) (G) |
| P2+PC | 0.00±0.0000(a) (A) | 2.40±0.0494(d) (B) | 3.85±0.0054(c) (C) | 4.88±0.0259(d) (D) | 6.14±0.0075(e) (E) | 6.60±0.0406(e) (F) | 7.89±0.0208(d) (G) |
| P5+PC | 0.00±0.0000(a) (A) | 2.20±0.0617(e) (B) | 3.55±0.0232(d) (C) | 4.61±0.0203(e) (D) | 5.58±0.0511(f) (E) | 6.40±0.0070(f) (F) | 7.39±0.0042(e) (G) |

C: control sample (without plasma treatment and phycocyanin pigment); PC-P: sample treated with phycocyanin pigment but without plasma; P2-PC: plasma-treated sample for 2 min without phycocyanin pigment; P5-PC: plasma-treated sample for 5 min without phycocyanin pigment; P2+PC: plasma-treated sample for 2 min with phycocyanin pigment; P5+PC: plasma-treated sample for 5 min with phycocyanin pigment. Different small and capital letters indicate significant differences in the columns and rows, respectively ($p < 0.05$). All data are expressed as mean±SEM (n=3). Data were analyzed using one-way ANOVA followed by Tukey's post hoc test ($p < 0.05$).

increased progressively in all groups, with the highest level observed in the control samples, reaching 10.28 log CFU/g on day 18 (p < 0.05). Samples treated with P5-PC and PC-P exhibited lower microbial proliferation, recording 8.22 and 9.83 log CFU/g, respectively (p < 0.05). The lowest bacterial counts were observed in the P5+PC samples (7.39 log CFU/g, p < 0.05).

### 3.3 Physicochemical properties

**3.3.1 PH and acidity.** As shown in Tables 5, no significant differences (p > 0.05) were observed among treatments on day 1. The pH of *Oncorhynchus mykiss* fillets increased progressively in all groups during 18 days of refrigerated storage (p < 0.05). The control samples exhibited the highest increase, rising from 6.38 on day 1 to 7.38 on day 18. In the P5-PC samples, pH increased from 6.39 to 7.07 by day 18, showing a significant difference compared with C (p < 0.05) but no significant difference from PC-P, which increased from 6.37 to 7.10 during the same period (p > 0.05). The P5+PC samples maintained the lowest pH values throughout storage, with a rise from 6.38 to 6.94 by day 18, and showed a significant difference from all other treatments at that time (p < 0.05).

As shown in S2 Table, the initial acidity value on day 1 was 0.13 in all samples, showing no significant differences (p > 0.05). Over the storage period, a gradual increase in acidity was observed in all groups. On day 18, the titratable acidity reached 0.23 in the control (C), while lower values were recorded in the treated samples: 0.19 in P5–PC, 0.20 in PC–P, and 0.17 in P5+PC, respectively. The lowest acidity increase was observed in the P5+PC group (p < 0.05), indicating that the combined treatment of gliding arc plasma and phycocyanin effectively delayed the acidification process in rainbow trout fillets during refrigerated storage.

**3.3.2 PV and TBARS.** As shown in Table 6, the initial peroxide value on day 1 was 0.21 in all samples, showing no significant differences (p > 0.05). During refrigerated storage, the peroxide value gradually increased in all groups. On day 18, the PV reached 0.54 in the control samples, while lower values were recorded in the treated samples: 0.47 in P5–PC, 0.35 in PC–P, and 0.33 in P5+PC, respectively. The differences among all groups on day 18 were statistically significant (p < 0.05). The lowest peroxide value observed in the P5+PC treatment indicated an effective inhibition of primary lipid oxidation during storage.

**Table 5. Mean PH of *Oncorhynchus mykiss* fillets treated with GAP and PCP during storage at 4°C for 18 days.**

| PH | Day1 | Day3 | Day6 | Day9 | Day12 | Day15 | Day18 |
|---|---|---|---|---|---|---|---|
| C | 6.38±0.0058(a) (A) | 6.57±0.0088(a) (B) | 6.71±0.0058(a) (C) | 7.00±0.0033(a) (D) | 7.13±0.0058(a) (E) | 7.27±0.0067(a) (F) | 7.38±0.0058(a) (G) |
| P2-PC | 6.38±0.0067(a) (A) | 6.48±0.0058(b) (B) | 6.66±0.0120(b) (C) | 6.85±0.0058(b) (D) | 7.02±0.0088(b) (E) | 7.07±0.0088(b) (F) | 7.16±0.0088(b) (G) |
| P5-PC | 6.39±0.0088(a) (A) | 6.43±0.0033(c) (B) | 6.58±0.0033(c) (C) | 6.80±0.0088(c) (D) | 6.93±0.0088(c) (E) | 7.01±0.0058(c) (F) | 7.07±0.0058(c) (G) |
| PC-P | 6.37±0.0000(a) (A) | 6.51±0.0067(d) (B) | 6.64±0.0033(b) (C) | 6.84±0.0058(b) (D) | 6.95±0.0088(c) (E) | 7.04±0.0033(c) (F) | 7.10±0.0120(c) (G) |
| P2+PC | 6.39±0.0000(a) (A) | 6.44±0.0058(c) (B) | 6.53±0.0058(d) (C) | 6.69±0.0088(d) (D) | 6.82±0.0058(d) (E) | 6.93±0.0033(d) (F) | 7.01±0.0058(d) (G) |
| P5+PC | 6.38±0.0088(a) (A) | 6.39±0.0033(e) (A) | 6.47±0.0033(e) (B) | 6.67±0.0088(d) (C) | 6.74±0.0058(e) (D) | 6.86±0.0058(e) (E) | 6.94±0.0058(e) (F) |

C: control sample (without plasma treatment and phycocyanin pigment); PC-P: sample treated with phycocyanin pigment but without plasma; P2-PC: plasma-treated sample for 2 min without phycocyanin pigment; P5-PC: plasma-treated sample for 5 min without phycocyanin pigment; P2+PC: plasma-treated sample for 2 min with phycocyanin pigment; P5+PC: plasma-treated sample for 5 min with phycocyanin pigment. Different small and capital letters indicate significant differences in the columns and rows, respectively (p < 0.05). All data are expressed as mean±SEM (n = 3). Data were analyzed using one-way ANOVA followed by Tukey's post hoc test (p < 0.05).

**Table 6. Mean PV of *Oncorhynchus mykiss* fillets treated with GAP and PCP during storage at 4°C for 18 days.**

| PV | Day1 | Day3 | Day6 | Day9 | Day12 | Day15 | Day18 |
|---|---|---|---|---|---|---|---|
| C | 0.21±0.0063(a) (A) | 0.27±0.0014(a) (B) | 0.30±0.0022(a) (C) | 0.34±0.0014(a) (D) | 0.40±0.0008(a) (E) | 0.46±0.0008(a) (F) | 0.54±0.0030(a) (G) |
| P2-PC | 0.21±0.0022(a) (A) | 0.21±0.0008(b) (A) | 0.26±0.0014(b) (B) | 0.30±0.0008(b) (C) | 0.33±0.0014(b) (D) | 0.40±0.0008(b) (E) | 0.43±0.0008(b) (F) |
| P5-PC | 0.21±0.0008(a) (A) | 0.24±0.0014(c) (B) | 0.28±0.0014(c) (C) | 0.31±0.0022(c) (D) | 0.37±0.0036(c) (E) | 0.42±0.0038(c) (F) | 0.47±0.0014(c) (G) |
| PC-P | 0.21±0.0030(a) (A) | 0.20±0.0017(d) (B) | 0.23±0.0008(d) (C) | 0.26±0.0014(d) (D) | 0.30±0.0008(d) (E) | 0.33±0.0008(d) (F) | 0.35±0.0014(d) (G) |
| P2+PC | 0.21±0.0014(a) (A) | 0.23±0.0014(e) (B) | 0.25±0.0008(b) (C) | 0.28±0.0014(e) (D) | 0.32±0.0014(e) (E) | 0.33±0.0038(e) (F) | 0.36±0.0008(e) (G) |
| P5+PC | 0.21±0.0014(a) (A) | 0.22±0.0008(b) (B) | 0.24±0.0014(d) (C) | 0.27±0.0014(d) (D) | 0.29±0.0008(d) (E) | 0.31±0.0022(d) (F) | 0.33±0.0014(f) (G) |

C: control sample (without plasma treatment and phycocyanin pigment); PC-P: sample treated with phycocyanin pigment but without plasma; P2-PC: plasma-treated sample for 2 min without phycocyanin pigment; P5-PC: plasma-treated sample for 5 min without phycocyanin pigment; P2+PC: plasma-treated sample for 2 min with phycocyanin pigment; P5+PC: plasma-treated sample for 5 min with phycocyanin pigment. Different small and capital letters indicate significant differences in the columns and rows, respectively (p<0.05). All data are expressed as mean±SEM (n=3). Data were analyzed using one-way ANOVA followed by Tukey's post hoc test (p<0.05).

As presented in Table S3, the TBARS values of *Oncorhynchus mykiss* fillets showed a significant (p<0.05) increase throughout 18 days of refrigerated storage. On day 1, significant differences (p<0.05) were already observed among samples, with initial values ranging from 0.16 to 0.21. The control samples exhibited the highest lipid oxidation, increasing from 0.21 on day 1 to 0.89 on day 18. In comparison, P5–PC showed a lower increase (0.19 to 0.60), while PC–P increased from 0.17 to 0.79. The P5+PC treatment exhibited the lowest TBARS values during storage, rising only from 0.16 to 0.51 by day 18. On day 18, all samples differed significantly (p<0.05), with the combined plasma–phycocyanin treatment (P5+PC) demonstrating the most effective inhibition of secondary lipid oxidation.

**3.3.3 TMA and TVN.** As shown in Table 7, TVN values of *Oncorhynchus mykiss* fillets increased significantly in all groups during 18 days of refrigerated storage (p<0.05). On day 1, no significant differences (p>0.05) were observed among treatments. In the C group, TVN values increased markedly from 8.77 mg N/100 g to 53.95 mg N/100 g by day 18, exceeding the acceptability limit of 30 mg N/100 g on day 9. The P5–PC treatment showed a lower increase, from 9.33 to 35.84 mg N/100 g, surpassing the limit on day 15. Similarly, the PC–P group increased from 9.71 to 48.35 mg N/100 g and also exceeded the threshold on day 15. In contrast, the P5+PC treatment exhibited the slowest accumulation of volatile nitrogen compounds, rising from 8.59 to only 29.40 mg N/100 g, and remained below the acceptability limit throughout the 18-day storage period (p<0.05).

As shown in S4 Table, TMA content of *Oncorhynchus mykiss* fillets increased progressively during refrigerated storage (p<0.05). On day 1, all samples exhibited similar TMA values (0.21 mg N/100 g) with no significant differences (p>0.05). By day 18, TMA levels increased in all groups, reaching 0.35 mg N/100 g in C, 0.24 mg N/100 g in P5–PC, 0.31 mg N/100 g in PC–P, and 0.21 mg N/100 g in P5+PC. The lowest TMA concentration observed in P5+PC indicated the most effective inhibition of nitrogenous compound formation during storage (p<0.05).

**3.3.4 Fatty acid profile (FAP).** Table 8 summarizes the fatty acid profile (FAP), showing 24 identified fatty acids categorized as saturated fatty acids (SFA), monounsaturated fatty acids (MUFA), and polyunsaturated fatty acids (PUFA).

According to Table 8, total SFA content varied significantly among the treatments (p<0.05). The highest total SFA value was recorded in P5–PC (22.55), followed by PC–P (18.19), while P5+PC (16.94) and C (16.10) showed the lowest levels. Among individual SFAs, palmitic acid (C16:0) was predominant across all groups, with the highest value observed in P5–PC (15.82) and the lowest in C (11.67). Stearic acid (C18:0) ranged from 2.84 in P5+PC to 4.64 in C, whereas myristic

**Table 7. Mean TVN of *Oncorhynchus mykiss* fillets treated with GAP and PCP during storage at 4°C for 18 days.**

| TVN | Day1 | Day3 | Day6 | Day9 | Day12 | Day15 | Day18 |
|---|---|---|---|---|---|---|---|
| C | 8.77±0.1867(a) (A) | 16.52±0.3233(a) (B) | 24.55±0.2469(a) (C) | 32.57±0.2469(a) (D) | 40.04±0.6466(a) (E) | 46.20±0.4850(a) (F) | 53.95±0.2469(a) (G) |
| P2-PC | 9.33±0.1867(a) (A) | 15.03±0.1867(b) (B) | 21.37±0.0764(b) (C) | 22.49±0.2469(b) (C) | 28.56±0.4277(b) (D) | 35.37±0.3365(b) (E) | 38.64±0.1617(b) (F) |
| P5-PC | 9.33±0.3733(a) (A) | 13.72±0.3233(c) (B) | 19.32±0.1316(bc) (C) | 22.96±0.1617(b) (D) | 25.95±0.2469(c) (E) | 32.85±0.0933(c) (F) | 35.84±0.4277(c) (G) |
| PC-P | 9.71±0.3733(a) (A) | 12.51±0.0933(d) (B) | 19.04±0.1617(bc) (C) | 26.04±0.1617(c) (D) | 31.73±0.0933(d) (E) | 39.29±0.1867(d) (F) | 48.35±0.3365(d) (G) |
| P2+PC | 9.33±0.3733(a) (A) | 11.67±0.1867(de) (B) | 18.20±0.1617(c) (C) | 21.84±0.1617(b) (D) | 27.72±0.1617(b) (E) | 32.20±0.1617(b) (F) | 35.28±0.3233(b) (G) |
| P5+PC | 8.59±0.1867(a) (A) | 11.11±0.1867(e) (B) | 16.33±0.3365(c) (C) | 20.25±0.4939(c) (D) | 23.80±0.1617(e) (E) | 27.63±0.2469(e) (F) | 29.40±0.1617(e) (G) |

C: control sample (without plasma treatment and phycocyanin pigment); PC-P: sample treated with phycocyanin pigment but without plasma; P2-PC: plasma-treated sample for 2 min without phycocyanin pigment; P5-PC: plasma-treated sample for 5 min without phycocyanin pigment; P2+PC: plasma-treated sample for 2 min with phycocyanin pigment; P5+PC: plasma-treated sample for 5 min with phycocyanin pigment. Different small and capital letters indicate significant differences in the columns and rows, respectively ($p < 0.05$). All data are expressed as mean±SEM ($n = 3$). Data were analyzed using one-way ANOVA followed by Tukey's post hoc test ($p < 0.05$).

acid (C14:0) varied from 0.79 (PC–P) to 1.14 (P5–PC). Pentadecanoic acid (C15:0) showed minimal variation among treatments, ranging from 0.16 to 0.21.

As shown in Table 8, total MUFA levels varied slightly among the treatments, ranging from 35.41 in P5+PC to 38.88 in C. Both P5-PC and P5+PC showed significant increases in MUFA levels compared with C ($p < 0.05$). Oleic acid (C18:1 n9c) was the dominant MUFA in all groups, with the highest value in C (35.36) and the lowest in P5-PC (30.31). Palmitoleic acid (C16:1) ranged from 0.21 in P5+PC to 4.10 in P5-PC, showing a notable rise in the P5-PC group. Minor MUFAs, such as gadoleic acid (C20:1) and heptadecanoic acid (C17:1), showed limited variation among treatments, with the latter peaking in P5-PC (3.32).

According to Table 8, the total PUFA content showed variations among treatments, ranging from 35.40 in C to 37.65 in P5+PC. Linoleic acid (C18:2 n6c) was the dominant PUFA, with values of 32.34, 33.78, 31.92, and 33.65 in C, PC-P, P5-PC, and P5+PC, respectively. γ-linolenic acid (C18:3 n6) increased from 1.17 in C to 2.84 in P5-PC and slightly decreased to 2.06 in P5+PC. Dihomo-γ-linolenic acid (C20:3 n6) remained nearly constant in C and PC-P (1.31, 1.31) but decreased in P5-PC (0.95) and increased again in P5+PC (1.34).

Arachidonic acid (C20:4 n6) showed a gradual elevation from 0.04 in C to 0.09 in P5+PC. Docosatetraenoic acid (C22:4 n6) and Dpan-6 (C22:5 n6) exhibited minor changes, with C22:4 n6 values of 0.20, 0.16, 0.16, and 0.20, and C22:5 n6 values of 0.34, 0.35, 0.25, and 0.31 across treatments.

Among omega-3 fatty acids, α-linolenic acid (C18:3 n3) decreased from 2.02 in C to 1.12 in P5-PC, while the P5+PC group showed a slight recovery (1.31). Eicosatrienoic acid (C20:3 n3) ranged from 0.96 in C to 1.08 in P5+PC. Eicosatetraenoic acid (C20:4 n3) increased gradually among treatments (0.18, 0.23, 0.29, 0.22). EPA (C20:5 n3) and DPA (C22:5 n3) exhibited limited variation, with EPA values of 0.19, 0.27, 0.19, and 0.20, and DPA values of 0.14, 0.14, 0.20, and 0.13. DHA (C22:6 n3) varied between 1.72 in C and 1.51 in P5+PC.

The total omega-6 content ranged from 35.40 in C to 37.65 in P5+PC, while total omega-3 content declined slightly from 5.21 in C to 4.45 in both P5-PC and P5+PC. The n-3/n-6 ratio decreased gradually across treatments, with values of 0.14, 0.13, 0.12, and 0.11 for C, PC-P, P5-PC, and P5+PC, respectively.

**3.3.5 Colorimetric analysis.** The *a** values of rainbow trout fillets increased significantly ($p < 0.05$) in all treatments during the 18-day storage period. In the control samples, redness rose from 2.26 on day 1 to 9.18 on day 18,

**Table 8. Fatty acid composition of *Oncorhynchus mykiss* fillets treated with GAP and PCP on day 1 of storage.**

| Fatty Acid | Name | C | PC-P | P5-PC | P5+PC |
|---|---|---|---|---|---|
| C14:0 | Myristic Acid | 0.81[a] | 0.79[b] | 1.14[c] | 0.93[d] |
| C15:0 | Pentadecanoic Acid | 0.21[a] | 0.20[b] | 0.21[a] | 0.16[c] |
| C16:0 | Palmitic Acid | 11.67[a] | 12.48[b] | 15.82[c] | 12.34[d] |
| C17:0 | Heptadecanoic Acid | 0.21[a] | 0.27[b] | 0.25[c] | 0.16[d] |
| C18:0 | Stearic Acid | 4.64[a] | 4.11[b] | 4.61[c] | 2.84[d] |
| C20:0 | Arachidic Acid | 0.44[a] | 0.34[b] | 0.52[c] | 0.51[d] |
| Saturated Fatty Acids[1] | | 16.1 | 18.19 | 22.55 | 16.94 |
| C16:1 | Palmitoleic Acid | 3.17[a] | 3.00[b] | 4.10[c] | 0.21[d] |
| C17:1 | | 0.16[a] | 0.12[b] | 3.32[c] | 0.13[d] |
| C18:1 n9c | Oleic Acid | 35.36[a] | 34.05[b] | 30.31[c] | 34.90[d] |
| C20:1 | Gadoleic Acid | 0.19[a] | 0.17[b] | 0.18[a] | 0.17[b] |
| Monounsaturated Fatty Acids[2] | | 38.88 | 37.34 | 37.91 | 35.41 |
| C18:2 n6c | LNA | 32.34[a] | 33.78[b] | 31.92[c] | 33.65[d] |
| C18:3 n6 | γ-linolenic Acid | 1.17[a] | 1.13[b] | 2.84[c] | 2.06[d] |
| C20:3 n6 | Dihomo-γ-linolenic Acid | 1.31[a] | 1.31[a] | 0.95[b] | 1.34[c] |
| C20:4 n6 | ARA | 0.04[a] | 0.05[b] | 0.07[c] | 0.09[d] |
| C22:4 n6 | DTA | 0.20[a] | 0.16[b] | 0.16[b] | 0.20[a] |
| C22:5 n6 | Dpan-6 | 0.34[a] | 0.35[a] | 0.25[b] | 0.31[c] |
| Omega 6 Fatty Acids[4] | | 35.40 | 36.78 | 36.19 | 37.65 |
| C18:3 n3 | ALA | 2.02[a] | 1.90[b] | 1.12[c] | 1.31[d] |
| C20:3 n3 | Eicosatrienoic Acid | 0.96[a] | 1.06[b] | 0.68[c] | 1.08[d] |
| C20:4 n3 | Eicosatetraenoic Acid | 0.18[a] | 0.23[b] | 0.29[c] | 0.22[d] |
| C20:5 n3 | EPA | 0.19[a] | 0.27[b] | 0.19[a] | 0.20[c] |
| C22:5 n3 | DPA | 0.14[a] | 0.14[a] | 0.20[b] | 0.13[c] |
| C22:6 n3 | DHA | 1.72[a] | 1.48[b] | 1.97[c] | 1.51[d] |
| Omega 3 Fatty Acids[3] | | 5.21 | 5.08 | 4.45 | 4.45 |
| C20:2 | Eicosadienoic acid (EDA) | 1.17[a] | 1.11[b] | 0.84[c] | 1.04[b] |
| C20:3 n9 | Mead Acid | 1.38[a] | 1.51[b] | 1.20[c] | 1.38[d] |
| n-3/n-6ratio[5] | | 0.14 | 0.13 | 0.12 | 0.11 |

C (control), PC–P (treated with *C*-phycocyanin followed by plasma), P5–PC (treated with plasma followed by *C*-phycocyanin), and P5+PC (combined treatment). Values are expressed as percentage of total fatty acids (mean ± SEM, n = 3). Different lowercase letters (a–d) in the same row indicate significant differences among samples (p < 0.05).

representing the largest increase among all samples. In contrast, P5-PC exhibited a moderate rise, increasing from 4.32 to 9.13, while PC-P and P5+PC showed higher initial redness levels that further intensified throughout storage. Specifically, *a** in PC-P increased from 8.64 to 12.68, and in P5+PC from 9.59 to 13.47, both displaying significant differences (p < 0.05) across all sampling days. Overall, the combined treatments (especially P5+PC) maintained higher *a** values throughout storage (S5 Table).

*b** values increased significantly (p < 0.05) throughout refrigerated storage in all treatments. The control samples showed a progressive increase from 2.08 on day 1 to 9.65 on day 18, with significant differences among storage days. In C2-PC and C5-PC, *b** values also increased gradually from 2.49 to 8.69 and from 3.01 to 7.98, respectively. Both groups showed significantly higher (p < 0.05) *b** values than C at the initial stage, but similar patterns of increase during storage. Phycocyanin-containing groups exhibited considerably higher *b** values throughout storage. In PC-P, *b** increased from 21.85 to 27.07; in P2+PC, from 23.39 to 28.00; and in P5+PC, from 24.22 to 27.96. At each sampling day, significant

Table 9. Mean ΔE of *Oncorhynchus mykiss* fillets treated with GAP and PCP during storage at 4°C for 18 days.

| ΔE | Day1 | Day3 | Day6 | Day9 | Day12 | Day15 | Day18 |
|---|---|---|---|---|---|---|---|
| C | 0.00±0.000(a) (A) | 0.68±0.1635(a) (B) | 2.09±0.0847(a) (C) | 3.72±0.0281(a) (D) | 6.89±0.2499(a) (E) | 9.44±0.0531(a) (F) | 13.26±0.0556(a) (G) |
| P2-PC | 0.00±0.0000(a) (A) | 0.80±0.0744(ab) (A) | 2.27±0.2114(a) (B) | 2.39±0.0857(bc) (B) | 5.60±0.2061(ab) (C) | 8.05±0.2867(b) (D) | 9.94±0.2215(b) (E) |
| P5-PC | 0.00±0.0000(a) (A) | 0.94±0.0805(ab) (B) | 1.89±0.0303(ab) (C) | 2.89±0.0677(bd) (D) | 4.40±0.0462(bde) (E) | 6.56±0.1109(c) (F) | 8.10±0.0677(c) (G) |
| PC-P | 0.00±0.0000(a) (A) | 0.97±0.2905(ab) (AB) | 1.67±0.2255(ab) (B) | 2.58±0.1722(bc) (B) | 5.02±0.6185(bd) (C) | 6.30±0.3633(c) (C) | 8.07±0.3550(c) (D) |
| P2+PC | 0.00±0.0000(a) (A) | 0.65±0.1105(a) (AB) | 1.22±0.1845(b) (B) | 2.13±0.1332(c) (C) | 3.50±0.2569(e) (D) | 6.02±0.0890(cd) (E) | 6.80±0.0758(d) (F) |
| P5+PC | 0.00±0.0000(a) (A) | 1.45±0.0396(b) (B) | 2.37±0.0407(a) (C) | 3.19±0.0851(d) (D) | 3.89±0.1199(de) (E) | 5.15±0.1294(d) (F) | 6.10±0.1336(d) (G) |

C: control sample (without plasma treatment and phycocyanin pigment); PC-P: sample treated with phycocyanin pigment but without plasma; P2-PC: plasma-treated sample for 2 min without phycocyanin pigment; P5-PC: plasma-treated sample for 5 min without phycocyanin pigment; P2+PC: plasma-treated sample for 2 min with phycocyanin pigment; P5+PC: plasma-treated sample for 5 min with phycocyanin pigment. Different small and capital letters indicate significant differences in the columns and rows, respectively (p<0.05). All data are expressed as mean±SEM (n=3). Data were analyzed using one-way ANOVA followed by Tukey's post hoc test (p<0.05).

(p<0.05) differences among treatments were observed. P5+PC generally exhibited the highest *b** values, followed by P2+PC and PC-P, while C had the lowest throughout the storage period (S6 Table ).

L* values of all samples decreased progressively throughout refrigerated storage (p<0.05). The control samples showed a continuous reduction from 70.23 on day 1 to 61.82 on day 18, representing the greatest overall decline. In contrast, samples treated with plasma and phycocyanin exhibited higher lightness stability. Specifically, C2-PC and C5-PC decreased from 69.62 to 63.77 and from 69.01 to 64.80, respectively. Among the individual treatments, PC-P declined from 67.16 to 62.55, while P2+PC decreased from 66.48 to 63.34. The smallest reduction was observed in P5+PC, which decreased from 65.55 to 62.73 by the end of storage. Statistical analysis indicated that differences among treatments were significant (p<0.05) during most sampling days, confirming that the combined plasma–phycocyanin treatment effectively preserved the lightness of trout fillets compared with the control (S7 Table).

As shown in Table 9, ΔE increased progressively throughout the 18-day refrigerated storage in all treatments (p<0.05). The control samples exhibited the largest overall change, with ΔE rising sharply from 0.00 on day 1 to 13.26 on day 18, reflecting substantial discoloration during storage. In contrast, all plasma- and phycocyanin-treated samples showed significantly smaller ΔE values, indicating improved color stability. Among the plasma-only groups, P2-PC and P5-PC increased from 0.00 to 9.94 and 8.10, respectively, while PC-P reached 8.07 by day 18. The combined treatments, particularly P2+PC and P5+PC, displayed the least total color change, with final ΔE values of 6.80 and 6.10, respectively. Statistical comparisons confirmed significant differences (p<0.05) among treatments at most time points, highlighting that the synergistic effect of GAP and PCP effectively minimized overall color deviation compared with the control.

### 3.3.6 Antioxidant evaluation. 3.3.6.1 DPPH

The DPPH radical scavenging activity of rainbow trout fillets during refrigerated storage is shown in Table 10. The control samples exhibited a gradual and significant increase (p<0.05) from 10.91 on day 1 to 36.41 on day 18. A similar but less pronounced increase was observed in P2-PC and P5-PC samples, which rose from 10.54 to 26.47 and 10.25 to 22.92, respectively, over the storage period. In contrast, the samples treated with PC-P, P2+PC, and P5+PC showed relatively stable DPPH values throughout storage. The PC-P samples increased slightly from 4.39 to 5.56, while P2+PC and P5+PC varied marginally from 4.29 to 5.38 and 4.20 to 5.23, respectively, without significant differences (p>0.05)

**Table 10. Mean DPPH of *Oncorhynchus mykiss* fillets treated with GAP and PCP during storage at 4°C for 18 days.**

| DPPH | Day1 | Day3 | Day6 | Day9 | Day12 | Day15 | Day18 |
|---|---|---|---|---|---|---|---|
| **C** | 10.91±0.2379(a) (A) | 13.75±0.7334(a) (AB) | 14.52±0.3176(a) (AB) | 16.01±0.8897(a) (B) | 20.84±0.9357(a) (C) | 24.12±0.5723(a) (C) | 36.41±0.4005(a) (D) |
| **P2-PC** | 10.54±0.0522(a) (A) | 11.86±0.1566(b) (A) | 14.14±0.2557(ab) (B) | 15.23±0.5962(a) (B) | 17.39±0.4455(b) (C) | 18.65±0.2521(b) (C) | 26.47±0.6710(b) (D) |
| **P5-PC** | 10.25±0.1681(a) (A) | 10.97±0.3719(b) (A) | 12.98±0.4236(b) (A) | 14.76±0.3601(a) (AB) | 16.17±0.1352(b) (BC) | 16.94±0.4632(c) (C) | 22.92±0.6795(c) (D) |
| **PC-P** | 4.39±0.3897(b) (ABC) | 3.99±0.1663(c) (A) | 4.35±0.1440(c) (AB) | 4.64±0.1664(b) (ABC) | 5.37±0.3060(c) (BC) | 5.21±0.2105(d) (BC) | 5.56±0.2529(d) (C) |
| **P2+PC** | 4.29±0.3774(b) (ABC) | 3.90±0.1533(c) (A) | 4.24±0.1382(c) (AB) | 4.52±0.1614(b) (ABC) | 5.22±0.2896(c) (BC) | 5.06±0.1951(d) (BC) | 5.38±0.2297(d) (C) |
| **P5+PC** | 4.20±0.3658(b) (AB) | 3.81±0.1668(c) (A) | 4.15±0.1468(c) (AB) | 4.41±0.1684(b) (AB) | 5.08±0.2786(c) (AB) | 4.92±0.2101(d) (B) | 5.23±0.2478(d) (B) |

C: control sample (without plasma treatment and phycocyanin pigment); PC-P: sample treated with phycocyanin pigment but without plasma; P2-PC: plasma-treated sample for 2 min without phycocyanin pigment; P5-PC: plasma-treated sample for 5 min without phycocyanin pigment; P2+PC: plasma-treated sample for 2 min with phycocyanin pigment; P5+PC: plasma-treated sample for 5 min with phycocyanin pigment. Different small and capital letters indicate significant differences in the columns and rows, respectively (p<0.05). All data are expressed as mean±SEM (n=3). Data were analyzed using one-way ANOVA followed by Tukey's post hoc test (p<0.05).

between day 1 and day 18. At the end of the storage period (day 18), the control samples showed significantly higher (p<0.05) DPPH values compared to all other treatments, except for P2-PC and P5-PC, which were statistically similar.

The ABTS radical scavenging activity of rainbow trout fillets during refrigerated storage is presented in S8 Table. The control samples showed a significant (p<0.05) increase from 4.44 on day 1 to 12.29 on day 18. Similarly, the P5-PC samples increased from 4.40 to 6.56 during the same period. At the beginning of storage (day 1), no significant differences (p>0.05) were observed among control samples, P2-PC, and P5-PC; however, by day 18, these groups showed significant differences (p<0.05). For PC-P, the ABTS activity increased slightly from 1.50 to 2.78, while the P5+PC samples rose from 1.39 to 2.30 by day 18. No significant differences (p>0.05) were observed among PC-P, P2+PC, and P5+PC either on day 1 or day 18 (S8 Table).

As shown in S9 Table, the FRAP values decreased significantly (p<0.05) in all groups during 18 days of refrigerated storage. The control samples decreased from 20.01 on day 1 to 11.75 on day 18, showing a significant reduction in reducing capacity. Similarly, the P5-PC samples declined from 21.00 to 15.20 during the same period. The PC-P group showed a reduction from 32.14 on day 1 to 18.46 on day 18, while the P5+PC samples decreased from 35.88 to 25.30. Overall, all treatments exhibited a gradual decline over time (p<0.05), though the extent of reduction varied among treatments.

**3.3.7 Sensory evaluation**

Odor scores decreased significantly (p<0.05) in all treatments throughout the 18 days of refrigerated storage. The control samples showed the greatest decline, decreasing from 5.00 on day 1 to 1.00 on day 18. In the P5-PC group, the odor score declined from 4.67 on day 1 to 1.67 on day 18, with no significant difference from the control samples at the beginning of storage. Similarly, the PC-P samples showed a reduction from 4.33 to 1.67, and the P5+PC samples decreased from 4.33 to 1.33 during the same period. Considering a sensory acceptability threshold of 3, the control samples crossed this limit by day 9, whereas all treated groups remained acceptable until day 12 (S2 Fig).

The texture scores of rainbow trout fillets gradually decreased during refrigerated storage in all treatments. In control samples, the texture values declined from 4.67 on day 1 to 1.67 on day 18, and considering the acceptance threshold of 3, these samples fell below the acceptable limit on day 6. The P5–PC samples exhibited the highest texture stability, with values decreasing from 5.00 on day 1 to 2.33 on day 18. Similarly, the PC–P group showed a reduction from 4.67 on day 1 to 2.67 on day 18. The P5+PC samples also decreased from 4.67 on day 1 to 2.67 on day 18, but they maintained the

acceptance threshold until day 9. Moreover, there were no significant differences among all samples on day 1, indicating that the treatments did not initially affect the textural quality of the fillets. In addition, in P5+PC samples, texture scores at day 9 and day 12 showed no significant differences, suggesting a more stable texture structure during mid-storage compared to other treatments (S3 Fig).

The color scores of all treatments gradually decreased during refrigerated storage. In control samples, the color values declined from 4.33 on day 1 to 1.00 on day 18, crossing the acceptance threshold on day 6. The P5–PC samples showed a similar decreasing trend, from 4.33 on day 1 to 1.33 on day 18. In the PC–P group, the color score decreased from 4.00 on day 1 to 2.33 on day 18, indicating better color retention compared to control samples. The P5+PC samples exhibited a decline from 4.00 on day 1 to 2.33 on day 18, maintaining the acceptance threshold until day 9. Furthermore, there was no significant difference between day 9 and day 10 in the P5+PC samples, demonstrating relatively stable color properties during mid-storage (S4 Fig).

The overall acceptability of all samples decreased gradually during refrigerated storage. In control samples, the mean score declined from 4.67 on day 1 to 1.33 on day 18, crossing the acceptance threshold (score = 3) as early as day 3, indicating a rapid loss of sensory quality. The P5–PC samples showed a slower decline, from 4.67 on day 1 to 2.00 on day 18, maintaining acceptable quality for a longer period. The PC–P group exhibited a decrease from 4.00 on day 1 to 2.33 on day 18, while the P5+PC samples decreased from 4.00 on day 1 to 2.33 on day 18, successfully maintaining the acceptance threshold until day 12. These results demonstrate that the combined treatments, particularly P5+PC, effectively preserved the sensory acceptability of the samples compared to control samples (Fig 3).

## 4. Discussion

In this study, two treatment durations (2 and 5 minutes) were applied to evaluate the effects of GAP, either alone or in combination with PCP (0.065 mg/Ml), on the shelf life and antioxidant capacity of rainbow trout fillets during refrigerated storage (4 °C) for 18 days. The results clearly demonstrated that GAP treatment alone effectively extended the shelf life compared with the control samples. Moreover, the combined GAP+PCP treatment exhibited a substantially stronger preservation effect than either treatment applied individually, highlighting a synergistic interaction between plasma-generated reactive species and the antioxidant activity of PCP.

The lower counts of TVC, PTC, *S. aureus*, *Lactobacillus*, and *Enterobacteriaceae* in the combined treatment samples compared to the control samples and GAP-treated samples may be attributed to enhanced penetration of PCP into bacterial cells through pores created by GAP treatment [16]. It has also been discovered that GAP alone yields superior results in microbial reduction compared to the control samples [26]. All microbiological analyses indicate that treating fish fillets between 2 and 5 minutes enhances their antimicrobial properties, and the results demonstrated that higher exposure times further improved the antimicrobial efficacy [27]. *Salmonella* spp. and various *E. coli* strains are among the most prevalent food pathogens and pose serious health threats. Under freezing conditions, *E. coli* can survive for up to 180 days [28]. In the current study, no *Salmonella* or *E. coli* were detected over the 18-day storage period, indicating effective inactivation. Although some studies have reported that plasma treatment had no significant effect on reducing *Lactobacillus* populations [28], the present findings revealed that GAP significantly decreased *Lactobacillus* counts, and that the combined GAP+PCP treatment further amplified this reduction. Furthermore, although previous studies reported complete inactivation of *Enterobacteriaceae* at exposure times longer than 5 minutes [29], in the present study a substantial reduction was achieved at a 5-minute treatment duration. Regarding shelf-life extension, the threshold for spoilage (TVC = 6 CFU/g) was surpassed in the control sample by day 6. However, the combined-treatment sample did not cross this spoilage limit until day 12 under a 5-minute exposure. This finding demonstrates the superiority of the combined treatment in delaying microbial spoilage. Overall, our study demonstrates that the combination of GAP and PCP creates a complementary effect, leading to enhanced antimicrobial efficacy. Phycocyanin, as an efficient electron donor, scavenges reactive radicals and interrupts their continuous generation, thereby mitigating oxidative stress induced by plasma

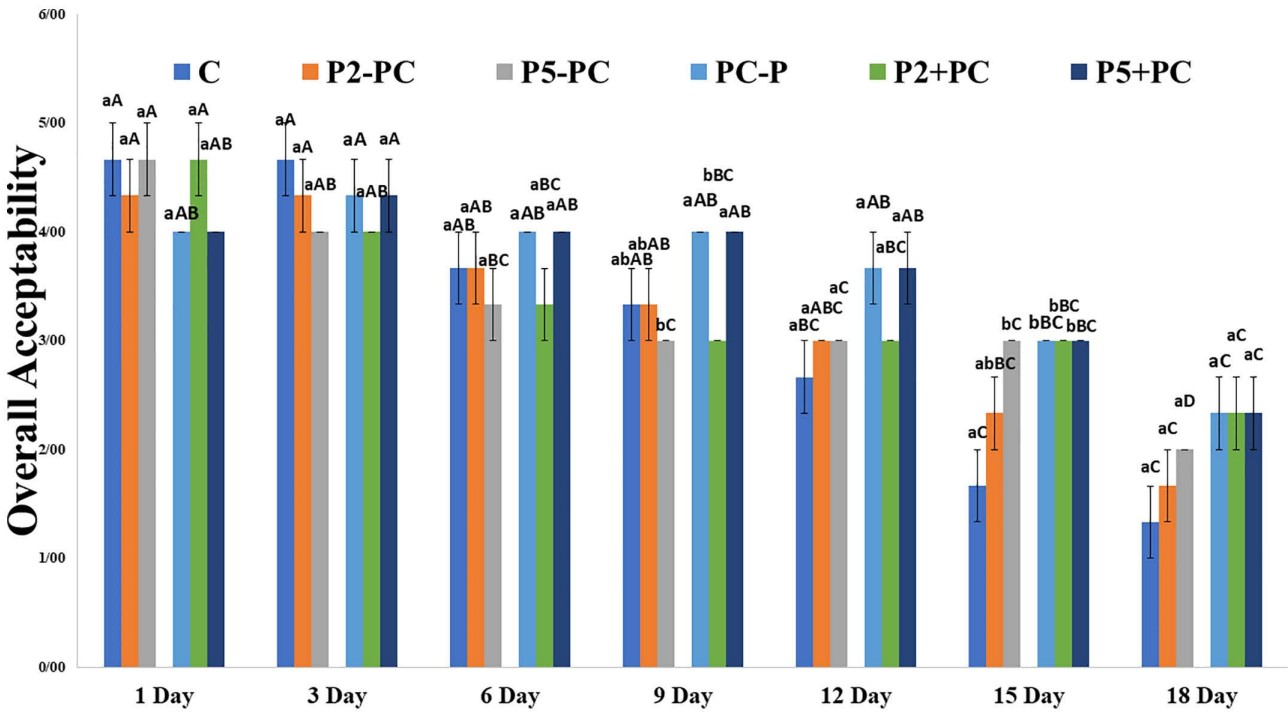

**Fig 3. Results of overall acceptability of rainbow trout fillets during 18 days of refrigerated storage (4 °C).** Sensory evaluation was performed on days 1, 3, 6, 9, 12, 15, and 18. Treatment groups: C, PC-P, P2-PC, P5-PC, P2+PC, and P5+PC. Values represent mean±SEM (n=3). Significant differences were determined by one-way ANOVA followed by Tukey's test (p<0.05). Different lowercase letters indicate significant differences among treatments within the same day, and uppercase letters among storage days within the same treatment.

treatment. This antioxidant and radical-trapping ability of PCP complements the oxidative mechanisms of GAP, resulting in a balanced yet powerful antimicrobial system. Thus, the combined GAP+PCP treatment offers a promising and innovative approach for microbial control and shelf-life extension in food products. Reactive species generated by GAP (including RONS, Ozone, atomic oxygen, UV radiation, photons, chemical reaction species, and charged particles), along with the phenolic compounds in PCP, can enter and damage cell membranes, leading to cell death [30,31]. Hydrophilic ROS (e.g., •OH, $HO_2$, and $H_2O_2$) generated by GAP amplify bactericidal action by disrupting membranes, whereas hydrophobic ROS contribute further oxidative stress [32]. Although the exact extent of UV contribution remains under debate, UV radiation generated within GAP can significantly enhance antimicrobial efficacy and disrupt membrane integrity; the resistance to UV-induced damage depends on the microorganism type [33].

The increase in pH observed during fish storage is primarily attributed to the breakdown of nucleotides and proteins, which leads to the accumulation of basic nitrogenous compounds [11]. In the present study, samples treated solely with PCP exhibited slightly higher pH values than those subjected to the combined GAP+PCP treatment. This trend may be linked to protein denaturation processes that reduce the number of acidic functional groups, thereby elevating the overall pH [34]. Interestingly, the PCP-treated fillets consistently displayed lower pH values compared to the control samples across all storage days, possibly due to the formation of nitrous ($HNO_2$) and nitric acids ($HNO_3$), as well as the dissociation of hydrogen ions ($H^+$) generated through plasma–phycocyanin interactions during treatment [34,35]. PV and TBARS are widely recognized as objective indicators of lipid oxidation, representing the formation of primary and secondary oxidation products, respectively. Lipase and phospholipase enzymes, particularly those associated with *Pseudomonas* species, catalyze the hydrolysis of lipids, increasing the concentration of free fatty acids that are highly susceptible to oxidation and

promote the formation of unstable lipid hydroperoxides [36]. Consequently, endogenous or microbial lipases contribute to elevated PV levels during storage. The observed inhibition of PV in samples treated with both GAP and PCP indicates a strong antioxidative synergy between the two treatments. Previous studies have reported that the effectiveness of plasma treatment largely depends on the type of carrier gas used [37]. In the present study, air was employed as the working gas, which proved to be highly effective while also being cost-efficient and environmentally friendly compared to noble or inert gases. Despite this, no statistically significant difference was observed in TBARS or PV values on the first day of storage following GAP exposure. Notably, the 5-minute treatment duration was more effective than the 2-minute exposure, as the combined GAP+PCP samples exhibited significantly lower TBARS values compared to both the control samples and GAP-only treatments. This effect can be attributed to the radical-scavenging and electron-donating capacity of PCP, which neutralizes RONS generated by GAP, thereby protecting lipids from oxidative degradation [38]. A TBARS value exceeding 2.0 mg MDA/kg is generally considered the threshold for fish spoilage, leading to deterioration in quality and flavor [39]. In the present study, all samples exhibited TBARS values well below this threshold after 18 days of refrigerated storage, confirming the high oxidative stability and overall quality of the fish fillets. Notably, the combined GAP+PCP-treated samples showed markedly lower TBARS values compared to the control samples, further demonstrating the superior protective effect of the combined treatment against lipid oxidation. TVN and TMA content were significantly lower in the GAP-treated group compared to the control samples ($p < 0.05$). Furthermore, the lowest TVN and TMA levels were observed in fillets subjected to the combined GAP+PCP treatment ($p < 0.05$), followed by those treated with PCP alone. The observed decline in microbial counts corresponded closely with the reduction in TVN and TMA, suggesting that the combined antimicrobial activity of GAP and the bioactive compounds in PCP effectively suppressed bacterial proliferation and deamination of non-protein nitrogenous compounds [40]. During prolonged storage, microorganisms proliferate and accelerate the breakdown of protein and non-protein nitrogenous compounds in fish tissue, leading to the formation of alkaline metabolites such as ammonia and various amines, which in turn increase the TVN content. The combined effect of PCP and GAP effectively slowed microbial growth and consequently limited the accumulation of these basic nitrogenous compounds, thereby controlling the rise in TVN values associated with protein degradation [41]. This finding suggests that the synergistic antimicrobial action of GAP and the bioactive components of PCP not only suppresses bacterial activity but also mitigates biochemical spoilage processes that compromise the sensory and nutritional quality of fish during refrigerated storage [11]. A total of 24 fatty acids were identified through analysis of the MUFA, PUFA, and SFA of C, PC-P, P5-PC, and P5+PC. As long as they are not oxidized, MUFA and PUFA are considered to be among the beneficial and necessary fats for human health. GAP and PCP treatments did not significantly change the composition of fatty acids. The fact that the TBARS index increased over the course of 18 days without a discernible change in the fatty acid composition indicates that non-lipid molecules such as proteins, nucleic acids, or carbohydrates can also form TBARS [42]. Due to the antioxidant activity of PCP, treatment with it and GAP can postpone the degradation of MUFA and PUFA [12]. SFAs are regarded as unhealthy fats. Among the SFAs that are most frequently found in plants and animals and that the human body can use to make energy is palmitic acid. On the other hand, excessive consumption may have negative health effects. Moreover, palmitic acid is the fatty acid found in fish in the greatest amount. Due to lipid oxidation caused by RONS generated by GAP, palmitic acid was found in the highest concentration in the P5-PC samples. However, P5+PC showed a lower amount than P5-PC, which is attributed to the presence of PCP and the reduced ROS production during GAP exposure due to the pigment's antioxidant properties [11,43,44]. The fish's L* and b* values were negatively affected by GAP treatment. The color of Fish flesh is influenced by the oxidation of lipids induced by RONS generated during GAP exposure, which damages pigment proteins in the tissue. Moreover, the color-forming molecules such as myoglobin (Mb) and hemoglobin (Hb) can undergo oxidative reactions due to RONS, resulting in darkening of the fish flesh. Yellowing of the fish during storage may also occur as a result of reactions between myoglobin and hydrogen peroxide [45]. Furthermore, amino acids and carbonyl-containing free radicals can participate in Maillard reactions, intensifying the yellowness of the fillets [46]. MDA formation often contributes to lowering the a*-value of fish flesh, a trend that aligns with TBARS

results [30]. PCP treatment exerted a greater influence on the a* and b* values than on the L* value, likely due to the blue hue of this pigment. During storage, the high-water activity and alkalinity of the samples, combined with the iron and copper content of the fish fillets, promoted a brownish tint. This browning is quantified by ΔE, where values of ΔE ≤ 1 typically indicate color differences barely perceptible to the human eye. The presence of active species generated by GAP and their interaction with the alkaline fish surface intensified browning when GAP was applied alone. However, combining GAP with PCP effectively counteracted these detrimental effects, likely owing to the strong antioxidant properties and blue coloration of PCP [38].

Fish fillets treated with GAP exhibited enhanced antioxidant activity with prolonged exposure (5 minutes). Consistent with previous studies, increasing treatment time generally enhanced the antioxidant potential of fish fillets [38]. Nevertheless, due to sample variability, the precise effect of GAP on antioxidant activity remains dependent on the specific conditions of treatment and sample composition. Furthermore, the remarkable synergistic effect of PCP and GAP in reducing DPPH and ABTS levels was highly significant; consequently, the P5+PC samples exhibited the lowest values for both parameters across all storage days. The marked reduction in DPPH and ABTS observed in the combined treatment compared with the GAP-treated and control samples can be attributed to the oxidation of ascorbic acid and the enhanced radical-scavenging activity of PCP [47]. The FRAP assay, commonly used to assess the electron-donating capacity of antioxidants, revealed a substantial increase in reducing power when GAP was combined with PCP. PCP prevented color deterioration in the fish fillets by inhibiting the conversion of $Fe^{3+}$/ ferricyanide to $Fe^{2+}$ and stabilizing the $Fe^{2+}$ complex [48].

Although GAP effectively inactivates many pathogens, certain RONS species generated during treatment may cause oxidative damage, leading to undesirable quality changes that shorten shelf life and reduce consumer acceptability. For example, partial inactivation of enzymes and microorganisms can induce discoloration (color loss or darkening), while lipid oxidation contributes to off-flavors and unpleasant odors that lower sensory scores. A clear correlation was observed between TVN and TBARS values and sensory attributes. Throughout the 18-day storage period, decreases in sensory scores corresponded with increases in TBARS and TVN values, reflecting ongoing lipid and protein oxidation. While GAP treatment alone contributed to this decline, the combined GAP+PCP treatment effectively mitigated these effects. The synergistic action of PCP minimized the adverse oxidative reactions caused by GAP, resulting in higher sensory scores compared to either GAP-only or PCP-only samples.Despite the gradual increase in TBARS and TVN during storage, all treated samples remained within acceptable sensory limits after 18 days, whereas the control (C) samples exhibited noticeable spoilage. The strong antioxidant potential of PCP—evidenced by increased FRAP values—also maintained favorable color stability and minimized oxidation-related discoloration in the pigmented fillets [49].

Overall, the combined treatment of GAP and PCP markedly enhanced the physicochemical, microbial, and sensory quality of rainbow trout fillets during refrigerated storage. This synergistic approach effectively suppressed lipid and protein oxidation, reduced spoilage indicators such as TVN and TMA, and significantly improved antioxidant capacity, as reflected by lower DPPH and ABTS values and higher FRAP activity. Furthermore, the incorporation of PCP mitigated plasma-induced oxidative stress, stabilized color parameters, and preserved desirable sensory attributes throughout storage. As a result, the shelf life of the treated fillets was successfully extended up to 12 days compared with the control samples. Collectively, these findings indicate that the GAP+PCP combination represents a sustainable, efficient, and cost-effective preservation strategy capable of maintaining the nutritional, visual, and microbial integrity of fish products while prolonging their marketability.

## 5. Conclusions

This study demonstrated that combining GAP (2 and 5 minutes) with PCP (0.065 mg/Ml), markedly enhanced the microbial stability, oxidative balance, fatty acid composition, and sensory quality of rainbow trout fillets during refrigerated storage at 4 °C. Based on the microbial spoilage threshold (TVC > 6 log CFU/g), the control samples reached unacceptable

levels after 3 days, whereas the GAP–PCP treatment (P5+PC) effectively extended the shelf life to 12 days. Although biochemical and sensory analyses corroborated this extended quality retention, microbial counts remained the most reliable indicator of spoilage, confirming 12 days as the practical shelf-life limit. Overall, the synergistic interaction between GAP-generated reactive oxygen species and the radical-scavenging, antioxidant properties of PCP provide a promising, non-thermal preservation approach for improving the safety, quality, and marketability of fresh fish products.

## Supporting information

**S1 Fig. Absorption spectrum of purified C-phycocyanin pigment extracted from cyanobacterial biomass.** The absorbance of the purified C-phycocyanin (PCP) was measured across the wavelength range of 260–800 nm, showing a distinct characteristic absorption peak at 621.9 nm, confirming the presence of highly pure PCP. The pronounced absorption peak at 621.9 nm indicates the presence of the chromophore phycocyanobilin and verifies the successful purification of the PCP.
(DOCX)

**S2 Fig. Results of odor of rainbow trout fillets during 18 days of refrigerated storage (4 °C).** Sensory evaluation was performed on days 1, 3, 6, 9, 12, 15, and 18. Treatment groups: C, PC-P, P2-PC, P5-PC, P2+PC, and P5+PC. Values represent mean±SEM (n=3). Significant differences were determined by one-way ANOVA followed by Tukey's test (p<0.05). Different lowercase letters indicate significant differences among treatments within the same day, and uppercase letters among storage days within the same treatment.
(DOCX)

**S3 Fig. Results of texture of rainbow trout fillets during 18 days of refrigerated storage (4 °C).** Sensory evaluation was performed on days 1, 3, 6, 9, 12, 15, and 18. Treatment groups: C, PC-P, P2-PC, P5-PC, P2+PC, and P5+PC. Values represent mean±SEM (n=3). Significant differences were determined by one-way ANOVA followed by Tukey's test (p<0.05). Different lowercase letters indicate significant differences among treatments within the same day, and uppercase letters among storage days within the same treatment.
(DOCX)

**S4 Fig. Results of color of rainbow trout fillets during 18 days of refrigerated storage (4 °C).** Sensory evaluation was performed on days 1, 3, 6, 9, 12, 15, and 18. Treatment groups: C, PC-P, P2-PC, P5-PC, P2+PC, and P5+PC. Values represent mean±SEM (n=3). Significant differences were determined by one-way ANOVA followed by Tukey's test (p<0.05). Different lowercase letters indicate significant differences among treatments within the same day, and uppercase letters among storage days within the same treatment.
(DOCX)

**S1 Table. Mean PTC of *Oncorhynchus mykiss* fillets treated with GAP and PCP during storage at 4°C for 18 days.** C: control sample (without plasma treatment and phycocyanin pigment); PC-P: sample treated with phycocyanin pigment but without plasma; P2-PC: plasma-treated sample for 2 min without phycocyanin pigment; P5-PC: plasma-treated sample for 5 min without phycocyanin pigment; P2+PC: plasma-treated sample for 2 min with phycocyanin pigment; P5+PC: plasma-treated sample for 5 min with phycocyanin pigment. Different small and capital letters indicate significant differences in the columns and rows, respectively (p<0.05). All data are expressed as mean±SEM (n=3). Data were analyzed using one-way ANOVA followed by Tukey's post hoc test (p<0.05).
(DOCX)

**S2 Table. Mean Acidity of *Oncorhynchus mykiss* fillets treated with GAP and PCP during storage at 4°C for 18 days.** C: control sample (without plasma treatment and phycocyanin pigment); PC-P: sample treated with phycocyanin

pigment but without plasma; P2-PC: plasma-treated sample for 2 min without phycocyanin pigment; P5-PC: plasma-treated sample for 5 min without phycocyanin pigment; P2+PC: plasma-treated sample for 2 min with phycocyanin pigment; P5+PC: plasma-treated sample for 5 min with phycocyanin pigment. Different small and capital letters indicate significant differences in the columns and rows, respectively (p<0.05). All data are expressed as mean±SEM (n=3). Data were analyzed using one-way ANOVA followed by Tukey's post hoc test (p<0.05).
(DOCX)

**S3 Table. Mean TBARS of *Oncorhynchus mykiss* fillets treated with GAP and PCP during storage at 4°C for 18 days.** C: control sample (without plasma treatment and phycocyanin pigment); PC-P: sample treated with phycocyanin pigment but without plasma; P2-PC: plasma-treated sample for 2 min without phycocyanin pigment; P5-PC: plasma-treated sample for 5 min without phycocyanin pigment; P2+PC: plasma-treated sample for 2 min with phycocyanin pigment; P5+PC: plasma-treated sample for 5 min with phycocyanin pigment. Different small and capital letters indicate significant differences in the columns and rows, respectively (p<0.05). All data are expressed as mean±SEM (n=3). Data were analyzed using one-way ANOVA followed by Tukey's post hoc test (p<0.05).
(DOCX)

**S4 Table. Mean TMA of *Oncorhynchus mykiss* fillets treated with GAP and PCP during storage at 4°C for 18 days.** C: control sample (without plasma treatment and phycocyanin pigment); PC-P: sample treated with phycocyanin pigment but without plasma; P2-PC: plasma-treated sample for 2 min without phycocyanin pigment; P5-PC: plasma-treated sample for 5 min without phycocyanin pigment; P2+PC: plasma-treated sample for 2 min with phycocyanin pigment; P5+PC: plasma-treated sample for 5 min with phycocyanin pigment. Different small and capital letters indicate significant differences in the columns and rows, respectively (p<0.05). All data are expressed as mean±SEM (n=3). Data were analyzed using one-way ANOVA followed by Tukey's post hoc test (p<0.05).
(DOCX)

**S5 Table. Mean a* of *Oncorhynchus mykiss* fillets treated with GAP and PCP during storage at 4°C for 18 days.** C: control sample (without plasma treatment and phycocyanin pigment); PC-P: sample treated with phycocyanin pigment but without plasma; P2-PC: plasma-treated sample for 2 min without phycocyanin pigment; P5-PC: plasma-treated sample for 5 min without phycocyanin pigment; P2+PC: plasma-treated sample for 2 min with phycocyanin pigment; P5+PC: plasma-treated sample for 5 min with phycocyanin pigment. Different small and capital letters indicate significant differences in the columns and rows, respectively (p<0.05). All data are expressed as mean±SEM (n=3). Data were analyzed using one-way ANOVA followed by Tukey's post hoc test (p<0.05).
(DOCX)

**S6 Table. . Mean b* of *Oncorhynchus mykiss* fillets treated with GAP and PCP during storage at 4°C for 18 days.** C: control sample (without plasma treatment and phycocyanin pigment); PC-P: sample treated with phycocyanin pigment but without plasma; P2-PC: plasma-treated sample for 2 min without phycocyanin pigment; P5-PC: plasma-treated sample for 5 min without phycocyanin pigment; P2+PC: plasma-treated sample for 2 min with phycocyanin pigment; P5+PC: plasma-treated sample for 5 min with phycocyanin pigment. Different small and capital letters indicate significant differences in the columns and rows, respectively (p<0.05). All data are expressed as mean±SEM (n=3). Data were analyzed using one-way ANOVA followed by Tukey's post hoc test (p<0.05).
(DOCX)

**S7 Table. Mean L* of *Oncorhynchus mykiss* fillets treated with GAP and PCP during storage at 4°C for 18 days.** C: control sample (without plasma treatment and phycocyanin pigment); PC-P: sample treated with phycocyanin pigment but without plasma; P2-PC: plasma-treated sample for 2 min without phycocyanin pigment; P5-PC: plasma-treated sample

for 5 min without phycocyanin pigment; P2+PC: plasma-treated sample for 2 min with phycocyanin pigment; P5+PC: plasma-treated sample for 5 min with phycocyanin pigment. Different small and capital letters indicate significant differences in the columns and rows, respectively (p < 0.05). All data are expressed as mean ± SEM (n = 3). Data were analyzed using one-way ANOVA followed by Tukey's post hoc test (p < 0.05).
(DOCX)

**S8 Table. Mean ABTS of *Oncorhynchus mykiss* fillets treated with GAP and PCP during storage at 4°C for 18 days.** C: control sample (without plasma treatment and phycocyanin pigment); PC-P: sample treated with phycocyanin pigment but without plasma; P2-PC: plasma-treated sample for 2 min without phycocyanin pigment; P5-PC: plasma-treated sample for 5 min without phycocyanin pigment; P2+PC: plasma-treated sample for 2 min with phycocyanin pigment; P5+PC: plasma-treated sample for 5 min with phycocyanin pigment. Different small and capital letters indicate significant differences in the columns and rows, respectively (p < 0.05). All data are expressed as mean ± SEM (n = 3). Data were analyzed using one-way ANOVA followed by Tukey's post hoc test (p < 0.05).
(DOCX)

**S9 Table. Mean FRAP of *Oncorhynchus mykiss* fillets treated with GAP and PCP during storage at 4°C for 18 days.** C: control sample (without plasma treatment and phycocyanin pigment); PC-P: sample treated with phycocyanin pigment but without plasma; P2-PC: plasma-treated sample for 2 min without phycocyanin pigment; P5-PC: plasma-treated sample for 5 min without phycocyanin pigment; P2+PC: plasma-treated sample for 2 min with phycocyanin pigment; P5+PC: plasma-treated sample for 5 min with phycocyanin pigment. Different small and capital letters indicate significant differences in the columns and rows, respectively (p < 0.05). All data are expressed as mean ± SEM (n = 3). Data were analyzed using one-way ANOVA followed by Tukey's post hoc test (p < 0.05).
(DOCX)

**S1 File. Supporting information.**
(XLSX)

## Author contributions

**Conceptualization:** Bahareh Nowruzi.

**Investigation:** Maedehsadat Seyedalangi.

**Methodology:** Maedehsadat Seyedalangi.

**Project administration:** Maedehsadat Seyedalangi, Bahareh Nowruzi.

**Resources:** Maedehsadat Seyedalangi, Seyed Amir Ali Anvar.

**Software:** Bahareh Nowruzi.

**Validation:** Amir Hossein Sari, Bahareh Nowruzi.

**Visualization:** Amir Hossein Sari, Bahareh Nowruzi.

**Writing – review & editing:** Bahareh Nowruzi.

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
