## [Decision Letter · Decision Letter 0]

8 Sep 2025

Dear Dr. Nowruzi,

Thank you for submitting your manuscript to PLOS ONE. After careful consideration, we feel that it has merit but does not fully meet PLOS ONE’s publication criteria as it currently stands. Therefore, we invite you to submit a revised version of the manuscript that addresses the points raised during the review process.

We look forward to receiving your revised manuscript.

Kind regards,

Amitava Mukherjee, ME, Ph.D.

Academic Editor

PLOS ONE

Journal Requirements:

3. Please upload a copy of Figure 2, to which you refer in your text on page 2. If the figure is no longer to be included as part of the submission please remove all reference to it within the text.

Reviewers' comments:

Reviewer's Responses to Questions

**Comments to the Author**

1. Is the manuscript technically sound, and do the data support the conclusions?

Reviewer #1: Yes

2. Has the statistical analysis been performed appropriately and rigorously?

Reviewer #1: Yes

3. Have the authors made all data underlying the findings in their manuscript fully available?

Reviewer #1: Yes

4. Is the manuscript presented in an intelligible fashion and written in standard English?

Reviewer #1: Yes

Reviewer #1: Your manuscript addresses a significant topic about food preservation with natural and non thermal technologies, however, the current version of the manuscript requires major revision:

Abstract and title:

The title needs grammatical correction and simplification.

The abstract should include specific numerical results, such as how much shelf life was extended or how antioxidant levels changed.

Grammar and English revision is required

Language:

The manuscript needs significant language editing. Many sentences are unclear, grammatically incorrect, or overly complex e.g., “According to[12]” Please consider having the paper reviewed by a fluent English-speaking editor.

Figures:

The figures are hard to read and overly complex. Consider adding a simple bar graphs with error pars to show the difference at different time points.

Methods and results:

1-The group labels (C, P2, PPC5, etc.) are confusing. Add a simple table early on that defines each group and explains what treatments they received.

2-Please explain why you chose 2 and 5 minutes for GAP and the specific concentration of PCP. Was this based on prior research or pilot testing?

3-Provide more detail about the 5-point hedonic scale used (e.g., what does a score of 3 mean?).

4-Report the mean scores with standard deviations or error bars. make a graphical representation that shows the results.

5-Indicate whether differences in sensory acceptability were statistically significant at each time point.

**Do you want your identity to be public for this peer review?** For information about this choice, including consent withdrawal, please see our Privacy Policy

Reviewer #1: No

---

## [Author Response · Author response to Decision Letter 1]

15 Oct 2025

Dear Prof. Amitava Mukherjee

PONE-D-25-22417

We would like to sincerely thank you and the reviewers for the time and effort dedicated to evaluating our manuscript entitled “Study on the Combined Effect of Gliding Arc Plasma Treatment and Phycocyanin Pigment on the Antimicrobial Activity of Rainbow Trout (Oncorhynchus mykiss)”.

We have carefully considered all comments and suggestions provided by the reviewers and the Academic Editor. We believe that these valuable critiques have greatly improved the overall quality and clarity of our manuscript. All changes made in the revised version are highlighted, and our detailed responses to each comment are provided below in bold. We hope that our revisions have satisfactorily addressed all concerns and that the manuscript is now suitable for publication in PLOS ONE.

Should you have any further questions or require additional information, please do not hesitate to contact me.

Sincerely yours,

Bahareh Nowruzi

Email: bahare77biol@gmail.com, baharehnowruzi77@iau.ac.ir

Academic Editor: Please ensure that your manuscript meets PLOS ONE's style requirements, including those for file naming. The PLOS ONE style templates can be found at

Authors: We sincerely thank the Academic Editor for this important reminder. We have carefully reviewed the PLOS ONE style templates and have reformatted our entire manuscript to ensure full compliance with all journal requirements. Specifically, we have:

1. Adjusted the main text layout, headings, and reference style to match the official PLOS ONE format.

2. Verified that all figures and tables are properly numbered, captioned, and cited in the text according to journal guidelines.

3. Updated file names for all figures, tables, and supporting information files following the required PLOS ONE naming conventions.

4. Checked that author names, affiliations, and corresponding author details are consistent with the PLOS ONE title page format.

We believe the revised manuscript now fully adheres to all PLOS ONE formatting and style standards.

Academic Editor: Your ethics statement should only appear in the Methods section of your manuscript. If your ethics statement is written in any section besides the Methods, please move it to the Methods section and delete it from any other section. Please ensure that your ethics statement is included in your manuscript, as the ethics statement entered into the online submission form will not be published alongside your manuscript.

Authors: We sincerely thank the Academic Editor for this important reminder. We have fully addressed this point. The ethics statement, which was previously located between the Conclusions and References sections, has been completely removed from that position and relocated to the Materials and Methods section (Page 10, Lines 203–229) of the revised manuscript. This version of the statement now serves as the sole and complete record of our ethical declaration in accordance with PLOS ONE policy.

Academic Editor: Please upload a copy of Figure 2, to which you refer in your text on page 2. If the figure is no longer to be included as part of the submission please remove all reference to it within the text.

Authors: We sincerely thank the Academic Editor for pointing out this issue. In the revised version of the manuscript, we have reconstructed Figure 2 and uploaded the updated version accordingly. Additionally, we have revised the sample names in the figure and throughout the text to ensure greater clarity and consistency.

Furthermore, a detailed explanation of the treatments and the corresponding sample labeling has been added to the Materials and Methods section (page 5, lines103-109) to facilitate better understanding.

Academic Editor: Please include captions for your Supporting Information files at the end of your manuscript, and update any in-text citations to match accordingly. Please see our Supporting Information guidelines for more information: http://journals.plos.org/plosone/s/supporting-information

Authors: We appreciate the Editor’s comment. We added all relevant data are within the paper and its Supporting Information files. S1 File. Raw data required to replicate all study findings reported in the article. (XLSX) in the declarations.

Academic Editor: If the reviewer comments include a recommendation to cite specific previously published works, please review and evaluate these publications to determine whether they are relevant and should be cited. There is no requirement to cite these works unless the editor has indicated otherwise.

Authors: We thank the Academic Editor for this clarification. Upon reviewing the reviewer’s comments, we noted that no specific previously published works were directly recommended for citation.

Academic Editor: 1. Is the manuscript technically sound, and do the data support the conclusions?

Reviewer #1: Yes

Authors: We sincerely thank the reviewer for acknowledging that the experiments were rigorously conducted and that the data appropriately support our conclusions.

Academic Editor: 2. Has the statistical analysis been performed appropriately and rigorously?

Reviewer #1: Yes

Authors: We appreciate the reviewer’s positive assessment regarding the appropriateness and rigor of our statistical analysis procedures.

Academic Editor: 3. Have the authors made all data underlying the findings in their manuscript fully available?

Reviewer #1: Yes

Authors: We sincerely thank the reviewer for confirming that all data underlying our findings have been made fully available in accordance with the PLOS ONE Data Availability Policy. All relevant data are included within the manuscript and its Supporting Information files.

Academic Editor: 4. Is the manuscript presented in an intelligible fashion and written in standard English?

Reviewer #1: Yes

Authors: We thank the reviewer for confirming that the manuscript is clearly presented and written in standard English. During the revision process, we further reviewed the entire text to correct minor grammatical and typographical errors and to ensure clarity and readability throughout the manuscript.

Reviewer #1: Your manuscript addresses a significant topic about food preservation with natural and non-thermal technologies, however, the current version of the manuscript requires major revision: Abstract and title:

The title needs grammatical correction and simplification.

Authors: We sincerely thank the reviewer for this valuable suggestion. We have revised the title to improve grammatical accuracy, clarity, and conciseness.

Reviewer #1: The abstract should include specific numerical results, such as how much shelf life was extended or how antioxidant levels changed.

Authors: We sincerely thank the reviewer for this insightful comment. In response, we have revised the Abstract to include specific numerical outcomes and quantitative results. The updated version now clearly reports the extent of shelf-life extension, microbial reduction, and antioxidant improvements achieved through the combined treatment of gliding-arc plasma (GAP) and phycocyanin pigment (PCP).

Reviewer #1: Grammar and English revision is required. Language:

The manuscript needs significant language editing. Many sentences are unclear, grammatically incorrect, or overly complex e.g., “According to[12]” Please consider having the paper reviewed by a fluent English-speaking editor.

Authors: We sincerely thank the reviewer for this valuable comment. In response, the entire manuscript has been thoroughly revised for English language, grammar, and overall clarity. We carefully reviewed all sentences to simplify overly complex phrasing and correct typographical and grammatical errors. Additionally, language editing support was provided by a fluent English-speaking scientific expert to ensure that the text meets the journal’s standards of clarity and readability. We believe these revisions have substantially improved the quality and readability of the manuscript.

Reviewer #1: Figures: The figures are hard to read and overly complex. Consider adding a simple bar graphs with error pars to show the difference at different time points.

Authors: We sincerely thank the reviewer for this valuable and constructive comment. In response, we have carefully revised all figures to enhance their clarity and scientific readability. Specifically, we:

• Replaced overly complex plots with simplified bar charts and line graphs displaying mean ± standard error of mean.

• Added error bars and statistical annotations (different letters or asterisks) to indicate significant differences (p < 0.05).

• Improved figure resolution and ensured all fonts and axis labels meet PLOS ONE’s formatting requirements.

• Updated figure captions to include details about statistical tests (two-way ANOVA and Tukey’s HSD post-hoc test) and sample sizes (n).

These modifications are reflected in figures 1–3 and figures S1-S4 of the revised manuscript.

We believe these changes have substantially improved the visual clarity and interpretability of our results in accordance with the reviewer’s valuable recommendations.

Reviewer #1: Methods and results: 1-The group labels (C, P2, PPC5, etc.) are confusing. Add a simple table early on that defines each group and explains what treatments they received.

Authors: We sincerely thank the reviewer for this helpful observation. To improve clarity and readability, we have added a detailed description of all experimental groups to the Materials and Methods section.

Specifically, we included a paragraph that defines each group abbreviation and a new fig 2 summarizing all treatment conditions. This addition appears in the revised manuscript on page 5 and 6, lines 103–128.

We believe that this new paragraph and table substantially improve the clarity of the study design.

Reviewer #1: 2-Please explain why you chose 2 and 5 minutes for GAP and the specific concentration of PCP. Was this based on prior research or pilot testing?

Authors: We sincerely thank the reviewer for this valuable comment. In our laboratory, a series of preliminary experiments were carried out to optimize plasma exposure time for rainbow trout fillets. These preliminary tests aimed to identify conditions that maximize microbial inactivation while minimizing any detrimental impact on product quality. During optimization, exposure durations shorter than 2 minutes produced insufficient levels of reactive oxygen and nitrogen species (RONS), resulting in limited antimicrobial activity. Conversely, treatment durations longer than 5 minutes led to visible surface drying, discoloration, and partial protein denaturation in the fillet tissue. Therefore, 2 minutes was selected as a lower exposure time to represent the rapid effect of plasma, and 5 minutes was chosen as an upper limit that allows the evaluation of cumulative plasma effects while maintaining product integrity.

For the phycocyanin pigment (PCP), the concentration of 0.065 mg/mL was selected based on antimicrobial screening and antioxidant stability tests. This level provided effective radical scavenging and microbial inhibition without altering the natural color or sensory attributes of the fish. Overall, these parameters represent an experimentally optimized balance between plasma efficacy, oxidative control, and product safety, ensuring reproducible and scientifically meaningful results.

Reviewer #1: 3-Provide more detail about the 5-point hedonic scale used (e.g., what does a score of 3 mean?).

Authors: We sincerely thank the reviewer for this important observation. In the revised manuscript, the sensory evaluation procedure has been described in greater detail to clarify the interpretation of the 5-point hedonic scale.

Specifically, the scale is now defined as follows: 1 = extremely good, 2 = good, 3 = acceptable, 4 = poor, and 5 = extremely unacceptable. A score of 3 thus indicates the threshold of acceptability, meaning that samples receiving this score were considered still acceptable for consumption but showing the onset of noticeable quality decline.

Reviewer #1: 4-Report the mean scores with standard deviations or error bars. make a graphical representation that shows the results.

Authors: We sincerely thank the reviewer for this valuable suggestion We would like to note that in our statistical analysis, the standard error of the mean (SEM) was used to represent data variability. Accordingly, all graphs have been updated to display error bars corresponding to SEM values for greater accuracy and consistency with our analytical approach.

Simplified bar graphs with SEM error bars have been prepared for the sensory parameters, including odor, texture, color, and overall acceptability. These revised figures provide a clearer visual representation of the sensory evaluation results, allowing for better understanding of the treatment effects and changes during refrigerated storage.

Reviewer #1: 5-Indicate whether differences in sensory acceptability were statistically significant at each time point.

Authors: We sincerely thank the reviewer for emphasizing this important point. In the revised manuscript, the sensory evaluation results have been comprehensively updated to clearly indicate the statistical significance of differences among treatments at each storage time.

All sensory data (colour, odour, texture, and overall acceptability) were analyzed using one-way ANOVA, and significant differences among mean values were determined using Tukey’s multiple comparison test at a significance level of p < 0.05. Distinct lowercase superscript letters within each column denote significant differences among treatments on the same storage day, while uppercase superscript letters indicate significant differences among storage days within the same treatment.

These statistical results are clearly described in the Results section (Pages 23–24, Lines 532–568).

---

## [Decision Letter · Decision Letter 1]

2 Nov 2025

Combined effects of gliding-arc plasma and C-phycocyanin on antioxidant activity and shelf-life extension of rainbow trout (Oncorhynchus mykiss) fillets

PONE-D-25-22417R1

Dear Dr. Nowruzi,

We’re pleased to inform you that your manuscript has been judged scientifically suitable for publication and will be formally accepted for publication once it meets all outstanding technical requirements.

Kind regards,

Amitava Mukherjee, ME, Ph.D.

Academic Editor

PLOS ONE

Additional Editor Comments (optional):

Reviewers' comments:

Reviewer's Responses to Questions

**Comments to the Author**

Reviewer #1: All comments have been addressed

2. Is the manuscript technically sound, and do the data support the conclusions?

Reviewer #1: Yes

3. Has the statistical analysis been performed appropriately and rigorously?

Reviewer #1: Yes

4. Have the authors made all data underlying the findings in their manuscript fully available?

Reviewer #1: Yes

5. Is the manuscript presented in an intelligible fashion and written in standard English?

Reviewer #1: Yes

Reviewer #1: (No Response)

**Do you want your identity to be public for this peer review?** For information about this choice, including consent withdrawal, please see our Privacy Policy

Reviewer #1: No

---

## [Editor Report · Acceptance letter]

PONE-D-25-22417R1

PLOS ONE

Dear Dr. Nowruzi,

I'm pleased to inform you that your manuscript has been deemed suitable for publication in PLOS ONE. Congratulations! Your manuscript is now being handed over to our production team.

Kind regards,

on behalf of

Professor Dr. Amitava Mukherjee

Academic Editor

PLOS ONE